# Smoking and alcoholism dual addiction dissemination model analysis with optimal control theory and cost-effectiveness

**Shewafera Wondimagegnhu Teklu**[1]*, **Belela Samuel Kotola**[2], **Haileyesus Tessema Alemneh**[3]

**1** Department of Mathematics, Natural and Computational Sciences, Debre Berhan University, Debre Birhan, Ethiopia, **2** Department of Mathematics, Natural and Computational Sciences, Oda Bultum University, Chiro, Ethiopia, **3** Department of Mathematics, Natural and Computational Sciences, University of Gondar, Gondar, Ethiopia

* luelzedo2008@gmail.com, shewaferaw@dbu.edu.et

**Data Availability Statement:** All relevant data are within the paper.

## Abstract

A mathematical model of the dual addiction dissemination dynamics of alcoholism and smoking was created and examined in this work, along with cost-effectiveness and optimal control techniques. The primary goal of the research is to determine which cost-efficient management techniques are most helpful in lowering the problem of dual addiction dispersion in the community. The smoking addiction sub-model, the alcohol addiction sub-model, and the dual addiction model between alcohol and smoking were all calculated, and their stability was examined in this study. The effective reproduction numbers of the models are computed using the next-generation operator technique. When the model's effective reproduction number is smaller than one, the backward bifurcation phenomenon is seen. Six time-dependent control measures are taken into consideration when formulating and analyzing the optimum control issue. Utilizing and applying the parameter values and using MATLAB ode45 solver we performed numerical simulations for both the dual addiction model and its optimal control problem. Furthermore, using the incremental cost-effectiveness ratio (ICER), we carried out the cost-effectiveness analyses. The cost-effectiveness analysis shows that implementing all the protection (education) control measures simultaneously (i.e., implementing Strategy A) is the most cost-effective strategy. Finally, we recommend that the public health stakeholders must put great effort into the implementation of Strategy A to reduce the smoking and alcoholism dual addiction dissemination problem in the community.

## 1. Introduction

Smoking is a major addiction problem caused by burning a substance, commonly tobacco [1, 2]. It is breathed through the mouth or the nose and enters directly into the lungs, where the active substance is moved from the lungs to the bloodstream and hence to the human brain [1, 2]. Different research studies proved that smoking has been one of the major health problems affecting various nations in the world, and of the more than seven million individuals who

**Funding:** The author(s) received no specific funding for this work.

**Competing interests:** The authors have declared that no competing interests exist.

died each year, more than six million were directly affected by smoking [3–5]. It is a major factor in various diseases like heart disease, cancer, stained teeth, stomach ulcers, vascular diseases, high blood pressure, and chronic obstructive lung diseases, among others [6–8]. Also, studies verified that in every year, nearly six million individuals die from smoking, and more than five million of these numbers are users and ex-users [4, 9]. The statistics related to death by smoking tobacco revealed that in every eight seconds there is at least one death, and 10% of adults who are smokers die of tobacco-related diseases. According to WHO prediction, 10 million individuals will die every year due to tobacco-related illnesses by 2030 [9].

Alcohol addiction (alcoholism), also known as alcohol dependence, is a social phenomenon characterized by an individual's uncontrollable desire for alcohol and potential physical dependence on it. It affects all social groups and individuals with different age groups and educational levels [10, 11]. It is a psychoactive substance with dependence-producing properties, and according to a World Health Organization (WHO) report globally, it is a cause for 3 million deaths per year, or almost 5.3% of all the deaths [11, 12]. The young generation can misuse alcohol for fun and enjoyment, but it affects many aspects of their lives [11, 13]. Also, various studies have demonstrated a high comorbidity between traditional cigarette smoking and high levels of alcohol consumption across a wide age demographic, including adolescents and young adults [14, 15]. It is very common for individuals to face problems with mental health and alcohol/drug use (co-occurring conditions) simultaneously, and studies reveal that the majority of drug (70%) and alcohol (86%) users in the population are affected by mental health problems [16]. Smoking is most common for individuals with both mental health conditions and those who use alcohol or drugs [16]. Excessive alcohol use and tobacco smoking are both serious public health problems [17]. Four and a half times as many adolescents who are smokers will face a higher risk of developing alcoholism addiction than never-smokers who drank a similar quantity of alcohol [17]. Consumption of alcohol and smoking tobacco have closely linked behaviors, and hence, not only are individuals who drink alcohol more likely to smoke (and vice versa), but also individuals who drink a large quantity of alcohol tend to smoke more cigarettes [15, 18]. More than 85 percent of adult individuals who drank alcohol are also smokers, and they may be more addicted to nicotine than smokers without a history of drinking [19]. One of the addictions is the cause of the other and their co-existence (dual addiction), and each of them is a trait for the health and economy of the community [19].

Mathematical modeling is a systematic approach that can be used to analyze the spread of infectious disease epidemics or the social behavior (interactions) of individuals using mathematical equations, functions, and relationships. Also, mathematical models are fundamental to research scholars for making predictions, analyzing behavior, and gaining insights without the need for costly or impractical experiments [20, 21]. In recent years, different researchers have constructed and analyzed mathematical models with integer order [22, 23] or fractional order [24–26] to investigate the dynamics of real-world situations such as infectious diseases, social phenomena, and population dynamics aspects. Mathematical models on the smoking addiction dissemination dynamics [5–7] and on the alcoholism dissemination dynamics [11–13] have been formulated and analyzed over the past few years. Analysis of optimal control problems with cost-effectiveness is a crucial tool in prioritizing the implementation of various possible intervention strategies in the prevention and control of infectious diseases and mental infections or addictions such as smoking addiction, alcohol addiction, and their dual addictions. It also provides the stakeholders and policymakers with important decision-making options on which prevention and control measures have the most cost-effective economic impact [27].

Researchers have studied different real-world situations using mathematical modeling approaches; for instance, Khyar et al. [1] developed and analyzed a mathematical model on

smoking with optimal control theory. The model numerical simulation results verify the stability of equilibrium points, confirm the theoretical findings, and reveal the role of optimal control strategy in controlling smoking severity. Sofia, I. R. et al. [2] formulated and analyzed a non-linear smoking model with optimal control strategies by incorporating media awareness and information. The study carried out a detailed qualitative analysis of the model and verified the results with quantitative (numerical simulation) results. The study also investigated the overall impact of media awareness on the achievement of smoking cessation. Ihsanjaya et al. [3] constructed and examined a change in smoking behavior integer order model that incorporates temporarily and permanently quitting smokers. The study performed sensitivity and numerical analysis and verified the qualitative analysis results with a quantitative approach, and the sensitivity analysis verified that a prevention strategy is better than quitting smoking. Verma and Vinay [5] formulated and examined a smoking model with an optimal control problem to investigate the impact of education and media awareness on the smokers' community. The results proved that in the absence and presence of media awareness, the smokers' community increases and decreases, respectively. Khajji et al. [10] constructed and analyzed a compartmental model on alcohol drinking to investigate the influence of alcohol treatment centers on the alcohol addiction dissemination phenomenon in the community. The study proved that the model equilibrium points are both locally and globally stable. Wang and Zejun [11] formulated and analyzed the alcoholism epidemic compartmental model with a saturated incidence rate and two distributed delays and investigated its global dynamics. From the results, it can be observed that individuals who are at the latent stage of alcoholism avoid excessive consumption of alcohol soon, and those who have recovered from treatment must avoid relapsing into excessive alcohol consumption in a short period of time. Shah et al. [28] formulated and analyzed a non-symmetric fractal-fractional model for ice smoking. The rigorous numerical simulation results reveal the applicability of the scheme, future prediction, and the effects of fractal-fractional orders simultaneously. Njagarah et al. [29] construct and analyze a model on the role of drug barons on the prevalence of drug epidemics in the population. Kotola and Teklu [30] constructed a racism and corruption co-existence model, similarly Teklu and Terefe [31] investigated the violence and racism co-existence dissemination as contagious disease dynamics in a community. Alemneh [32] constructed and analyzed a corruption model with optimal control strategies, and similarly, Alemneh et al. [22] formulated and analyzed a mathematical model on social media addiction dissemination in the community with optimal control strategies. Asamoah, Joshua Kiddy, et al. [33] investigate the impacts of optimal control strategies on the spreading dynamics of gonorrhea in a structured population. The study considered intervention measures such as educating individuals about the effects of gonorrhea infection spreading, the use of condoms during sexual activity, gonorrhea vaccination, and gonorrhea treatment for both male and female populations. The results of the study reveal that implementing all the proposed control measures has the optimal effect of reducing the gonorrhea infection impact on both the male and female populations. Makinde, O. D. et al. [34] formulated and analyzed a mathematical model to investigate the impact of drug abuse on a nation's education sector; Abidemi, A., and J. O. Akanni [35] constructed and analyzed a mathematical model to investigate the impact of illicit drug use and banditry population with optimal control strategies and cost-effectiveness. From the cost-effectiveness analysis results, they proved that a strategy that combines all four control intervention measures is the most effective and cost-effective strategy to recommend for public health stakeholders. Akanni, J. O., et al. [36] developed and analyzed the financial crime population dynamics with optimal control measures and cost-effectiveness. The researchers proposed two time-dependent optimal control measures: the public enlightenment campaign (preventive) and corrective measures on the financial crime dynamics in the community. The cost-effectiveness investigation

proved that implementation of the optimal enlightenment campaign must be intensified to prevent unsuspecting susceptible (naive) individuals from being influenced by acts of financial crime, and Bhunu, Claver P., and Steady Mushayabasa [37] formulated and theoretically analyzed the smoking and alcoholism dual usage compartmental model. Their model did not consider optimal control theory and cost-effectiveness analysis. The simulation results proved that encouraging and supporting all smokers (alcohol drinkers) to quit smoking (alcohol drinking) also contributes meaningfully to alcohol (smoking) control programs.

We confirmed that no mathematical model researchers on alcohol addiction, smoking addiction, or alcohol and smoking dual addiction included the permanently dual addicted group, the protected group (protected compartment) against both addictions (through education), optimal control theory, and cost-effectiveness analysis in the dynamics of these addictions' spread, despite the fact that some researchers formulated and analyzed the smoking and alcoholism dual existence dynamics. Similarly, no scientific investigation, including mathematical modeling of drinking and smoking, looked at the phenomenon of backward bifurcation. In light of this, we are driven to develop and evaluate the dual addiction model between drinking and smoking in this study, which includes six time-dependent control measures and cost-effectiveness. Investigating the most economical, ideal control approach is the primary objective. The dual addiction model between alcoholism and smoking explained the co-existing characteristics of both addictions with time-dependent control strategies. A detailed qualitative (mathematical) analysis of the dual addiction model is presented. The Incremental Cost-Effectiveness Ratio (ICER) approach is used to perform cost-effectiveness analyses. Graphical representations of the suggested control strategies combined in various cases are presented, and the results are compared. These are the most significant contributions of this study. The remaining parts of this study are: Section 2 of this study consists of the construction of the dual addiction model and the proof of positivity and boundedness of its solutions; Section 3 conducts a qualitative analysis of the sub-models and the dual addiction model and sensitivity analysis; Section 4 re-formulates and analyzes the optimal control problem; Section carry out the numerical simulations; section 6 perform the cost-effectiveness analysis; and Section 7 discussed the conclusion of the study and gives future directions.

## 2. Dual addiction integer order model formulation

In this work, we developed a dual addiction model between alcoholism and smoking by using similar model assumption methodologies as in previous works [38–41]. We did this by splitting the whole human population, represented by N(t), into eleven different groups, including individuals who do not consume alcohol, do not smoke but they are at risk for either smoking or alcoholism addiction denoted by $S(t)$, individuals who are protected against either smoking addiction (by education) or alcohol addiction (by education) denoted by $P(t)$, individuals exposed to smoking denoted by $E_S(t)$, individuals who are addicted with smoking, but, can be improved (rehabilitated by taking treatment) denoted by $I_S(t)$, individuals who are permanently addicted with smoking throughout their life (i.e., could not be rehabilitated by taking any treatment measures) denoted by $P_S(t)$, individuals who are exposed to alcohol addiction denoted by $E_A(t)$, individuals who are addicted with alcohol, but, can be improved (rehabilitated by taking treatment) denoted by $I_A(t)$, individuals who are permanently addicted with alcohol throughout their life (i.e., could not be rehabilitated by taking any treatment measures) denoted by $P_A(t)$, individuals who are dually addicted with smoking and alcohol denoted by $C_{SA}(t)$, individuals who are permanently dual addicted with both smoking and alcohol denoted by $P_{SA}(t)$, and individuals who are improved (rehabilitated) from either smoking or alcohol or

dual addiction denoted by $T(t)$, such that

$$N(t) = S(t) + P(t) + E_S(t) + I_S(t) + P_S(t) + E_A(t) + I_A(t) + P_A(t) + C_{SA}(t) + P_{SA}(t) \\ + T(t). \tag{1}$$

Let's assume the incidence rates as follows, using criteria similar to those used to research co-infections with infectious diseases, such as [40]. People who are at risk of alcohol addiction develop alcohol addiction at the conventional incidence rate provided by

$$\lambda_A(t) = \frac{\beta_1}{N(t)}(I_A(t) + P_A(t) + \theta_1(C_{SA}(t) + P_{SA}(t))), \tag{2}$$

Where $\theta_1 \geq 1$ is the modification parameter that increases dissemination rate and $\beta_1$ is the alcohol addiction dissemination rate, and similarly, individuals who are at risk for smoking addiction acquire smoking addiction at the standard incidence rate given by

$$\lambda_S(t) = \frac{\beta_2}{N(t)}(I_S(t) + P_S(t) + \theta_2(C_{SA}(t) + P_{SA}(t))), \tag{3}$$

Where $\theta_2 \geq 1$ is the modification parameter that increases smoking dissemination rate and $\beta_2$ is the smoking addiction dissemination rate.

Following the model assumption methodologies described in the works [39, 40, 42], we take into consideration the following core presumptions:

- $\pi$ is portion of the recruited individuals who are entered to the protected group,

- Individuals who are at risk for both smoking and alcoholism are increased by the portion $(1 - \pi)$ of the recruitment rate,

- The human population is homogeneous in every group,

- Individuals in each group are subject to natural death rate $\mu$,

- Human population is variable,

- Individuals will be permanent smokers or permanent alcohol drinkers or permanently dual addicted,

- Individuals who are either smoking or alcohol permanently addicted will be dually addicted before permanently dual addicted,

- The control measures considered are education, punishment, and psychological treatments,

- Permanently addicted individuals are individuals who are addicted throughout their life, i.e., they will not be rehabilitated through any control measures.

Using the model assumptions, descriptions of the state variables and parameters in Tables 1 and 2 respectively, the schematic diagram for the smoking and alcoholism dual addiction dissemination dynamics is given in Fig 1.

Using the schematic diagram given by Fig 1 above, the integer order non-linear ordinary differential equations governed by the descriptions and the assumptions are

**Table 1. Description of the model parameters.**

| Parameter | Description |
|---|---|
| $\mu$ | Natural death rate |
| K | Individuals recruitment rate |
| $\pi$ | Portion of recruited individuals entered to the protection group |
| $\delta_1 < 1$ | Modification parameter that shows protected individuals have less possibility to be exposed to smoking addiction than susceptible individuals |
| $0 < \delta_2 < \delta_3 > 1$ | Modification parameters that show alcohol addicted individuals have high possibility to smoking addiction than susceptible individuals |
| $0 < \rho_1 < 1$ | Modification parameter that shows protected individuals have less possibility to be exposed to alcohol addiction than susceptible individuals |
| $0 < \rho_2 > \rho_3 > 1$ | Modification parameter that shows smoking addicted individuals have high possibility to alcohol addiction than susceptible individuals |
| $\varepsilon_1$ | Improvement rate of alcohol addicted population |
| $\varepsilon_2$ | Improvement rate of smoking addicted population |
| $\varepsilon_3$ | Improvement of dually addicted individuals with smoking and alcohol |
| $\beta_1$ | Alcohol addiction dissemination rate |
| $\beta_2$ | Smoking addiction dissemination rate |
| $\alpha_2$ | Progression rate from alcohol addicted to alcohol permanently addicted group |
| $\gamma_1$ | Progression rate from smoking exposed to smoking addicted group |
| $\alpha_1$ | Progression rate from alcohol exposed to alcohol addicted group |
| $\delta$ | The rate of protection of individuals who are at risk for both addictions |
| $\alpha$ | Progression rate from dually addicted to dual permanent addicted group |
| $\gamma_2$ | Progression rate from smoking addicted to smoking permanently addicted group |
| $\theta_1$ | The modification parameter that verify smoking and alcohol dual addicted individuals are more involved in the alcohol addiction dissemination process than alcohol only addicted individuals |
| $\theta_2$ | The modification parameter that verify smoking and alcohol dual addicted individuals are more involved in the smoking addiction dissemination process than smoking only addicted individuals |
| $d_S$ | Death rate due to smoking addiction related illness |
| $d_A$ | Death rate due to alcohol addiction related illness |
| $d_{PS}$ | Death rate due to permanent smoking addiction related illness |
| $d_{PA}$ | Death rate due to permanent alcohol addiction related illness |
| $d_C$ | Death rate due to smoking and alcohol dual addiction related illness |
| $d_{sA}$ | Death rate due to smoking and alcohol permanent dual addiction related illness |

**Table 2. Descriptions of the model state variables.**

| Variable | Descriptions |
|---|---|
| $S$ | Individuals who are at risk to both smoking and alcohol addiction |
| $P$ | Individuals protected by education against both smoking and alcohol addictions |
| $E_S$ | Individuals exposed to smoking |
| $I_S$ | Individuals who are addicted with smoking |
| $P_S$ | Permanently smoker individuals |
| $E_A$ | Individuals exposed to alcohol addiction |
| $I_A$ | Individuals who are addicted with alcohol |
| $P_A$ | Individuals who are permanently addicted with alcohol |
| $C_{SA}$ | Individuals who are dually addicted with both smoking and alcohol |
| $P_{SA}$ | Individuals who are dually permanently addicted with both smoking and alcohol |
| $T$ | Individuals who improved smoking addiction or alcohol addiction or dual addiction |

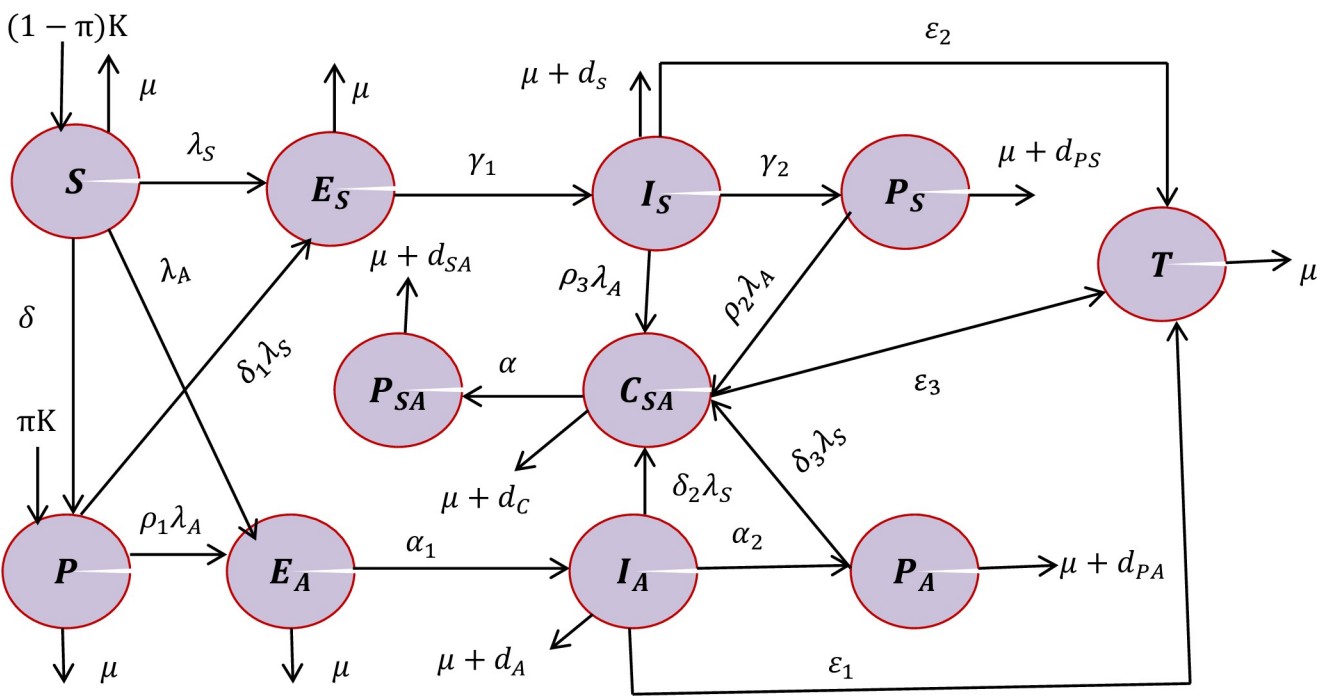

**Fig 1.** The schematic diagram for alcohol and smoking dual addiction dissemination dynamics in a community where $\lambda_A(t)$ and $\lambda_S(t)$ are stated in (2) and (3) respectively.

represented by:

$$\frac{dS}{dt} = (1-\pi)\mathrm{K} - (\lambda_S + \lambda_A + \delta + \mu)S,$$

$$\frac{dP}{dt} = \pi\mathrm{K} + \delta S - (\rho_1\lambda_A + \delta_1\lambda_S + \mu)P,$$

$$\frac{dE_S}{dt} = \lambda_S S + \delta_1\lambda_S P - (\mu + \gamma_1)E_S, \tag{4}$$

$$\frac{dI_S}{dt} = \gamma_1 E_S - (\varepsilon_2 + \gamma_2 + \rho_3\lambda_A + \mu + d_S)I_S,$$

$$\frac{dP_S}{dt} = \gamma_2 I_S - (\rho_2\lambda_A + \mu + d_{PS})P_S,$$

$$\frac{dE_A}{dt} = \lambda_A S + \rho_1\lambda_A P - (\mu + \alpha_1)E_A,$$

$$\frac{dI_A}{dt} = \alpha_1 E_A - (\alpha_2 + \varepsilon_1 + \delta_2\lambda_S + \mu + d_A)I_A,$$

$$\frac{dP_A}{dt} = \alpha_2 I_A - (\delta_3\lambda_S + \mu + d_{PA})P_A,$$

$$\frac{dC_{SA}}{dt} = \delta_2\lambda_S I_A + \delta_3\lambda_S P_A + \rho_3\lambda_A I_S + \rho_2\lambda_A P_S - (\alpha + \varepsilon_3 + \mu + d_C)C_{SA},$$

$$\frac{dP_{SA}}{dt} = \alpha C_{SA} - (\mu + d_{SA})P_{SA},$$

$$\frac{dT}{dt} = \varepsilon_3 C_{SA} + \varepsilon_1 I_A + \varepsilon_2 I_S - \mu T,$$

with population initial data given by

$$S(0) > 0, P(0) \geq 0, E_S(0) \geq 0, I_S(0) \geq 0, P_S(0) \geq 0, E_A(0) \geq 0, I_A(0) \geq 0, P_A(0) \geq 0, C_{SA}(0) \geq 0, P_{SA}(0) \geq 0, \text{ and } T(0) \geq 0. \tag{5}$$

The sum of all the differential equations described in Eq (4) is represented by

$$\frac{dN}{dt} = \mathrm{K} - \mu N - d_S I_S - d_{PS} P_S - d_A I_A - d_{PA} P_A - d_C C_{SA} - d_{SA} P_{SA}.$$

## 2.1 Basic properties of the dual addiction model (4)

The smoking and alcohol dual addiction dissemination model described in (4) is well-posed if and only if all the model solutions (state variables) are non-negative and bounded in the invariant region

$$\Omega = \left\{ (S, P, E_S, I_S, P_S, E_A, I_A, P_A, C_{SA}, P_{SA}, T) \in \mathbb{R}_+^{11}, N(t) \leq \frac{\mathrm{K}}{\mu} \right\}. \tag{6}$$

**Theorem 1 (Positivity of the Model Solutions):** Given the initial data as stated in Eq (5), let's look at the solutions $S(t)$, $P(t)$, $E_S(t)$, $I_S(t)$, $P_S(t)$, $E_A(t)$, $I_A(t)$, $P_A(t)$, $C_{SA}(t)$, $P_{SA}(t)$, and $T(t)$ of the smoking and alcohol dual addiction dissemination model (4) are non-negative for all time $t>0$.

**Proof:** Let us consider $S(0)>0$, $P(0)>0$, $E_S(0)>0$, $I_S(0)>0$, $P_S(0)>0$, $E_A(0)>0$, $I_A(0)>0$, $P_A(0)>0$, $C_{SA}(0)>0$, $P_{SA}(0)>0$, and $T(0)>0$ then for all t $>$ 0, we have to show that $S(t)>0$, $P(t)>0$, $E_S(t)>0$, $I_S(t)>0$, $P_S(t)>0$, $E_A(t)>0$, $I_A(t)>0$, $P_A(t)>0$, $C_{SA}(t)>0$, $P_{SA}(t)>0$, and $T(t)>0$.

Utilizing the identical methodology employed to establish positivity of the model solutions in [4, 6–8], let us define $\tau = \sup\{t>0: S(t)>0, P(t)>0, E_S(t)>0, I_S(t)>0, P_S(t)>0, E_A(t)>0, I_A(t)>0, P_A(t)>0, C_{SA}(t)>0, P_{SA}(t)>0, \text{ and } T(t)>0\}$. Now since all the smoking and alcohol dual addiction model state variables

$S(t), P(t), E_S(t), I_S(t), P_S(t), E_A(t), I_A(t), P_A(t), C_{SA}(t), P_{SA}(t)$, and $T(t)$ are continuous we can justify that $\tau>0$. If $\tau = +\infty$, then positivity holds. But, if $0 < \tau < +\infty$ we will have $S(\tau) = 0, P(\tau) = 0, E_S(\tau) = 0, I_S(\tau) = 0, P_S(\tau) = 0, E_A(\tau) = 0, I_A(\tau) = 0, P_A(\tau) = 0, C_{SA}(\tau) = 0, P_A(\tau) = 0,$ and $T(\tau) = 0$. From the first equation of the smoking and alcohol dual addiction dissemination model (4) we have determined that

$$\frac{dS}{dt} + (\lambda_S + \lambda_A + \delta + \mu)S = (1 - \pi)\mathrm{K},$$

and integrate using method of integrating factor we have determined the constant value

$S(\tau) = M_1 S(0) + M_1 \int_0^\tau exp^{\int(\lambda_S(t)+\lambda_A(t)+\delta+\mu)dt}(1 - \pi)\mathrm{K}dt > 0$ where $M_1 = exp^{-(\mu\tau+\delta\tau+\int_0^\tau(\lambda_S(w)+\lambda_A(w)))} > 0$, $S(0)>0$, and the exponential function always is positive, then the solution $S(\tau) >0$ hence $S(\tau)\neq0$.

Again from the second equation of the model (4) we have determined that
$\frac{dP}{dt} + (\rho_1 \lambda_A + \delta_1 \lambda_S + \mu)P = \pi K + \delta S$, and also using the method of integrating factor after
some calculations we got $P(\tau) = M_1 P(0) + M_1 \int_0^\tau exp^{\int (\rho_1 \lambda_A + \delta_1 \lambda_S + \mu))dt}(\pi K + \delta S)dt > 0$ where
$M_1 = exp^{-(\mu\tau + \int_0^\tau (\rho_1 \lambda_A(w) + \delta_1 \lambda_S(w))} > 0$, $P(0) > 0$, and from the meaning of $\tau$, the solution $P(\tau) > 0$
hence $P(\tau) \neq 0$.

Similarly, $E_S(\tau) > 0$ hence $E_S(\tau) \neq 0$, $I_S(\tau) > 0$ hence $I_S(\tau) \neq 0$. $P_A(\tau) > 0$ hence $P_A(\tau) \neq 0$,
$E_A(\tau) > 0$ hence $E_A(\tau) \neq 0$, $I_A(\tau) > 0$ hence $I_A(\tau) \neq 0$, $P_A(\tau) > 0$ hence $P_A(\tau) \neq 0$, $C_{SA}(\tau) > 0$
hence $C_{SA}(\tau) \neq 0$, $P_{SA}(\tau) > 0$ hence $P_{SA}(\tau) \neq 0$, and $T(\tau) > 0$ hence $T(\tau) \neq 0$. Thus, $\tau = +\infty$, and
hence all the solutions of the smoking and alcohol dual addiction dissemination system (4) are
non-negative.

**Theorem 2 (The Invariant Region):** All the solutions of the smoking and alcohol dual
addiction dissemination model (4) are bounded in the region stated in Eq (6).

**Proof:** Let $(S, P, E_S, I_S, P_S, E_A, I_A, P_A, C_{SA}, P_{SA}, T) \in \mathbb{R}_+^{11}$ be an arbitrary non-negative solu-
tion of the system (4) with initial conditions given in Eq (5). Now adding all the differential
equations given in Eq (4) to determine the derivative of the total population N given by $\frac{dN}{dt} =$
$K - \mu N(t) - d_s I_S(t) - d_{PS} P_S(t) - d_A I_A(t) - d_{PA} P_A(t) - d_C C_{SA}(t) - d_{SA} P_{SA}(t)$. Then, based on
the proof of Theorem 1 above we have the result $\frac{dN}{dt} \leq K - \mu N(t)$ and using the Theorem of
Birkhoff's and Rota's whenever $t \to \infty$, we have obtained that $0 \leq N(t) \leq \frac{K}{\mu}$. Thus, all of the via-
ble and positive solutions for the dual addiction dissemination model (4) of alcohol and smok-
ing entered in the region specified by Eq (6).

**Note:** The model (4) solutions are both positive and bounded in the region $\Omega =$
$\left\{ (S, P, E_S, I_S, P_S, E_A, I_A, P_A, C_{SA}, P_{SA}, T) \in \mathbb{R}_+^{11}, N(t) \leq \frac{K}{\mu} \right\}$ which is invariant and attracting for
dual addiction dynamical system (4). Therefore, the dual addiction model (4) is both mathe-
matically and epidemiologically well-posed by taking into account the epidemic behaviors of
drunkenness and smoking addictions, and it is sufficient to take into account the dynamics of
the flow created by the system (4) in $\Omega$ [23, 42].

## 3. The models' qualitative analyses

It is necessary to acquire a fundamental understanding of the alcohol addiction and smoking
addiction dissemination sub-models prior to delving into an analysis of the Eq (4) dual addic-
tion dissemination model.

## 3.1 Sub-model for alcohol addiction dissemination

This subsection makes the assumption that there isn't a persistent smoking addiction in the
general public, i.e. $E_S = I_S = P_S = C_{SA} = P_{SA} = 0$ and the set of ordinary differential equations in
model (4) represents the alcohol addiction dissemination sub-model as follows:

$$\frac{dS}{dt} = (1-\pi)K - (\lambda_A + \delta + \mu)S,$$

$$\frac{dP}{dt} = \pi K + \delta S - (\rho_1 \lambda_A + \mu)P,$$

$$\frac{dE_A}{dt} = \lambda_A S + \rho_1 \lambda_A P - (\mu + \alpha_1)E_A, \tag{7}$$

$$\frac{dI_A}{dt} = \alpha_1 E_A - (\alpha_2 + \varepsilon_1 + \mu + d_A)I_A,$$

$$\frac{dP_A}{dt} = \alpha_2 I_A - (\mu + d_{PA})P_A,$$

$$\frac{dT}{dt} = \varepsilon_1 I_A - \mu T,$$

where the total population is $N_1(t) = S(t) + P(t) + E_A(t) + I_A(t) + P_A(t) + T(t)$, the alcohol addiction dissemination sub-model force of infection is given by $\lambda_A = \frac{\beta_1}{N_1}(I_A(t) + P_A(t))$ and with initial data $S(0){>}0, P(0){\geq}0, E_A(0){\geq}0, I_A(0){\geq}0, P_A(0){\geq}0$ and $T(0){\geq}0$. In a way comparable to the dual addiction diffusion model for alcohol and smoking outlined in (4) within the region $\Omega_1 = \left\{ (S + P + E_A + I_A + P_A + T) \in \mathbb{R}^6_+, N_1(t) \leq \frac{K}{\mu} \right\}$, it is sufficient to consider the dynamics of the sub-model (7) in $\Omega_1$ is well-posed.

**3.1.1 Alcohol addiction-free equilibrium point and its local stability.** Setting the addicted state variables and setting the right-hand side of the sub-model (7) to zero and computing the results yields the alcohol addiction free equilibrium point given by
$E_A^0 = (S^0, P^0, 0, 0, 0, 0) = \left( \frac{(1-\pi)K}{\delta+\mu}, \frac{\delta K + K\pi\mu}{\mu(\delta+\mu)}, 0, 0, 0, 0 \right)$.

Using the next generation operator approach discovered by van den Driesch and Warmouth [43], the effective reproduction number of the alcohol addiction free equilibrium point, represented by $\mathcal{R}_{0A}$, is used to analyze the local stability of the equilibrium point. Using this technique, we were able to calculate the dissemination matrix F and the transition matrix V, which are provided by

$$F = \begin{pmatrix} 0 & \frac{\beta_1}{N_1}(S^0 + \rho_1 P^0) & \frac{\beta_1}{N_1}(S^0 + \rho_1 P^0) & 0 \\ 0 & 0 & 0 & 0 \\ 0 & 0 & 0 & 0 \\ 0 & 0 & 0 & 0 \end{pmatrix},$$

and

$$V = \begin{pmatrix} (\mu + \alpha_1) & 0 & 0 & 0 \\ -\alpha_1 & (\alpha_2 + \varepsilon_1 + \mu + d_A) & 0 & 0 \\ 0 & -\alpha_2 & \mu + d_{PA} & 0 \\ 0 & -\varepsilon_1 & 0 & \mu \end{pmatrix}.$$

Afterwards, we have calculated the result provided by using Mathematica as

$$V^{-1} = \begin{pmatrix} \frac{1}{(\mu + \alpha_1)} & 0 & 0 & 0 \\ \frac{\alpha_1}{(\mu + \alpha_1)(\alpha_2 + \varepsilon_1 + \mu + d_A)} & \frac{1}{(\alpha_2 + \varepsilon_1 + \mu + d_A)} & 0 & 0 \\ \frac{\alpha_1 \alpha_2}{\mu(\mu + \alpha_1)(\alpha_2 + \varepsilon_1 + \mu + d_A)} & \frac{\alpha_2}{\mu(\alpha_2 + \varepsilon_1 + \mu + d_A)} & \frac{1}{\mu + d_{PA}} & 0 \\ \frac{\alpha_1 \varepsilon_1}{\mu(\mu + \alpha_1)(\alpha_2 + \varepsilon_1 + \mu + d_A)} & \frac{\varepsilon_1}{\mu(\alpha_2 + \varepsilon_1 + \mu + d_A)} & 0 & \frac{1}{\mu} \end{pmatrix},$$

and

$$FV^{-1} = \begin{pmatrix} \dfrac{\beta_1\alpha_1(\mu+\alpha_2)(S^0+\rho_1 P^0)}{N_1(\mu+\alpha_1)(\alpha_2+\varepsilon_1+\mu+d_A)(\mu+\delta)} & \dfrac{\beta_1[\mu S^0+\alpha_2\rho_1 P^0]}{N_1\mu(\alpha_2+\varepsilon_1+\mu+d_A)} & \dfrac{\beta_1\rho_1 P^0}{N_1(\mu+d_{PA})} & 0 \\ 0 & 0 & 0 & 0 \\ 0 & 0 & 0 & 0 \\ 0 & 0 & 0 & 0 \end{pmatrix}.$$

Next, the greatest eigenvalue in magnitude of the next generation matrix $FV^{-1}$ is equal to the effective reproduction number of the alcohol addiction sub-model (7) given by

$$\mathcal{R}_{0A} = \frac{\beta_1\alpha_1\mu(\mu+\alpha_2)(1-\pi)}{\mu(\mu+\alpha_1)(\alpha_2+\varepsilon_1+\mu+d_A)(\delta+\mu)} + \frac{\beta_1\alpha_1\rho_1(\mu+\alpha_2)(\delta+\pi\mu)}{\mu(\mu+\alpha_1)(\alpha_2+\varepsilon_1+\mu+d_A)(\delta+\mu)}.$$ Therefore, the average number of secondary alcohol addiction protected and susceptible cases obtained from a typical permanent alcohol addiction or alcohol addiction formed individual during his/her effective addiction time in a susceptible population is described as $\mathcal{R}_{0A}$ in this context. Hence, the threshold result, $\mathcal{R}_{0A}$, represents the effective reproduction number for the alcohol addiction sub-model.

**Theorem 3:** The alcohol addiction free equilibrium point $E_A^0$ of the dynamical system (7) is locally asymptotically stable (LAS) if $\mathcal{R}_{0A}<1$, and it is unstable if $\mathcal{R}_{0A}>1$.

**Proof:** Using the Routh-Hurwitz stability criteria from [44], the local stability of the alcohol addiction free equilibrium point of the alcohol addiction sub-model (7) is examined. At the alcohol addiction free equilibrium point $E_A^0$, the Jacobian matrix of the alcohol addiction sub-model (7) is as follows:

$$J(E_A^0) = \begin{bmatrix} -(\delta+\mu) & 0 & 0 & -\dfrac{\beta_1 S^0}{S^0+P^0} & -\dfrac{\beta_1 S^0}{S^0+P^0} & 0 \\ \delta & -\mu & 0 & -\dfrac{\beta_1\rho_1 P^0}{S^0+P^0} & -\dfrac{\beta_1\rho_1 P^0}{S^0+P^0} & 0 \\ 0 & 0 & -(\mu+\alpha_1) & \dfrac{\beta_1 S^0}{S^0+P^0}+\dfrac{\beta_1\rho_1 P^0}{S^0+P^0} & \dfrac{\beta_1 S^0}{S^0+P^0}+\dfrac{\beta_1\rho_1 P^0}{S^0+P^0} & 0 \\ 0 & 0 & \alpha_1 & -(\alpha_2+\varepsilon_1+\mu+d_A) & 0 & 0 \\ 0 & 0 & 0 & \alpha_2 & -\mu-d_{PA} & 0 \\ 0 & 0 & 0 & \varepsilon_1 & 0 & -\mu \end{bmatrix}.$$

Then, the corresponding characteristic equation of the Jacobian matrix $J(E_A^0)$ is given by

$$\begin{vmatrix} -(\delta+\mu)-\lambda & 0 & 0 & -\dfrac{\beta_1 S^0}{S^0+P^0} & -\dfrac{\beta_1 S^0}{S^0+P^0} & 0 \\ \delta & -\mu-\lambda & 0 & -\dfrac{\beta_1\rho_1 P^0}{S^0+P^0} & -\dfrac{\beta_1\rho_1 P^0}{S^0+P^0} & 0 \\ 0 & 0 & -(\mu+\alpha_1)-\lambda & \dfrac{\beta_1(S^0+\rho_1 P^0)}{S^0+P^0} & \dfrac{\beta_1(S^0+\rho_1 P^0)}{S^0+P^0} & 0 \\ 0 & 0 & \alpha_1 & -(\alpha_2+\varepsilon_1+\mu+d_A)-\lambda & 0 & 0 \\ 0 & 0 & 0 & \alpha_2 & -\mu-d_{PA}-\lambda & 0 \\ 0 & 0 & 0 & \varepsilon_1 & 0 & -\mu-\lambda \end{vmatrix}$$
$$= 0.$$

Finally, we have determined the results given by $\lambda = -\mu < 0$, or $\lambda = -(\delta + \mu) < 0$, or $\lambda = -\mu < 0$, or

$$\lambda^3 + a\lambda^2 + b\lambda + c = 0, \tag{8}$$

where

$$a = \mu + (\mu + \alpha_1) + (\alpha_2 + \varepsilon_1 + \mu) > 0,$$

$$b = \mu((\mu + \alpha_1) + (\alpha_2 + \varepsilon_1 + \mu)) + (\mu + \alpha_1)(\alpha_2 + \varepsilon_1 + \mu + d_A) - \frac{\beta_1\alpha_1(S^0 + \rho_1 P^0)}{S^0 + P^0},$$

and

$$c = \left( (\mu + d_{PA})(\mu + \alpha_1)(\alpha_2 + \mathcal{A}_5 + \mu) - \frac{\beta_1\alpha_1(S^0 + \rho_1 P^0)(\mu + \alpha_2)}{S^0 + P^0} \right)$$

$$= (\mu + d_{PA})(\mu + \alpha_1)(\alpha_2 + \mathcal{A}_5 + \mu)$$

$$\times \left( 1 - \frac{\beta_1\alpha_1\mu(\mu + \alpha_2)(1 - \pi)}{\mu(\mu + \alpha_1)(\alpha_2 + \varepsilon_1 + \mu + d_A)(\delta + \mu)} + \frac{\beta_1\alpha_1\rho_1(\mu + \alpha_2)(\delta + \pi\mu)}{\mu(\mu + \alpha_1)(\alpha_2 + \varepsilon_1 + \mu + d_A)(\delta + \mu)} \right)$$

$$= \mu + d_{PA})(\mu + \alpha_1)(\alpha_2 + \mathcal{A}_5 + \mu)(1 - \mathcal{R}_{0A}) < 0 \text{ iff } \mathcal{R}_{0A} < 1.$$

Then, using the cubic polynomial Eq (8), we applied the Routh-Hurwitz stability criteria to determine that each eigenvalue is negative if and only if $\mathcal{R}_{0A} < 1$. Since all of the eigenvalues are negative whenever $\mathcal{R}_{0A} < 1$, we can therefore conclude that the model's alcohol addiction free equilibrium point is locally asymptotically stable whenever $\mathcal{R}_{0A} < 1$. From Lemma 1, biologically speaking, we can conclude that if the initial size of the sub-populations of the alcohol addiction sub-model given in Eq (7) is in the basin of attraction of the alcohol addiction free equilibrium point $E_A^0$, then the alcohol addiction dissemination dynamics can be eliminated from the population (whenever $\mathcal{R}_{0A} < 1$).

**3.1.2 Existence of alcohol addiction dominance equilibrium point(s).** Let $E_A^* = (S^*, P^*, E_A^*, I_A^*, P_A^*, T^*)$ be an arbitrary alcohol addiction dominance equilibrium point of the alcohol addiction sub-model (7) and is determined by making the right hand side of Eq (7) as zero. Then, after a number of steps of computations we have got

$$S^* = \frac{(1 - \pi)K}{(\lambda_A^* + \delta + \mu)}, P^* = \frac{\pi K(\lambda_A^* + \delta + \mu) + (1 - \pi)K\delta}{(\lambda_A^* + \delta + \mu)(\rho_1\lambda_A^* + \mu)},$$

$$E_A^* = \frac{(1 - \pi)K\lambda_A^*(\rho_1\lambda_A^* + \mu + \delta\rho_1) + \pi K\rho_1\lambda_A^*(\lambda_A^* + \delta + \mu)}{(\lambda_A^* + \delta + \mu)(\rho_1\lambda_A^* + \mu)(\mu + \alpha_1)},$$

$$I_A^* = \frac{\alpha_1[(1 - \pi)K\lambda_A^*(\rho_1\lambda_A^* + \mu + \delta\rho_1) + \pi K\rho_1\lambda_A^*(\lambda_A^* + \delta + \mu)]}{(\lambda_A^* + \delta + \mu)(\rho_1\lambda_A^* + \mu)(\mu + \alpha_1)(\alpha_2 + \varepsilon_1 + \mu + d_A)}, \tag{9}$$

$$P_A^* = \frac{\alpha_1\alpha_2[(1 - \pi)K\lambda_A^*(\rho_1\lambda_A^* + \mu + \delta\rho_1) + \pi K\rho_1\lambda_A^*(\lambda_A^* + \delta + \mu)]}{(\mu + d_{PA})(\lambda_A^* + \delta + \mu)(\rho_1\lambda_A^* + \mu)(\mu + \alpha_1)(\alpha_2 + \varepsilon_1 + \mu + d_A)},$$

$$T^* = \frac{\alpha_1\varepsilon_1[(1 - \pi)K\lambda_A^*(\rho_1\lambda_A^* + \mu + \delta\rho_1) + \pi K\rho_1\lambda_A^*(\lambda_A^* + \delta + \mu)]}{(\mu + d_{PA})(\lambda_A^* + \delta + \mu)(\rho_1\lambda_A^* + \mu)(\mu + \alpha_2)(\alpha_2 + \varepsilon_1 + \mu + d_A)},$$

Now substitute $I_A^*$ and $P_A^*$ given in Eq (9) in to

$\lambda_A^* = \frac{\beta_1 I_A^* + \beta_1 P_A^*}{S^* + P^* + \mathcal{B}_S^* + I_A^* + P_A^* + T^*}$, then after some simplifications we have

$$D_3(\lambda_M^*)^2 + D_2\lambda_M^* + D_1 = 0, \tag{10}$$

where

$$D_3 = \frac{(\mu(\alpha_2 + \varepsilon_1 + \mu + d_A) + \alpha_1(\mu + \varepsilon_1))\mu K\rho_1 + K\rho_1\alpha_1\alpha_2\mu}{\mu} > 0,$$

$$D_2 = \frac{(\mu(\alpha_2 + \varepsilon_1 + \mu) + \alpha_1(\mu + \varepsilon_1))\mu + \alpha_1\alpha_2\mu}{\mu + d_{P_A}} + \frac{(\mu + \alpha_1)(\alpha_2 + \varepsilon_1 + \mu + d_A)\mu(\pi + \rho_1 - \pi\rho_1) - \beta_1\alpha_1(\mu + \alpha_2)\rho_1}{(\mu - \pi\mu + \delta\rho_1 + \pi\mu\rho_1)},$$

and

$$D_1 = K(\mu + \alpha_1)(\alpha_2 + \varepsilon_1 + \mu + d_A)(\mu + d_{P_A})(\delta + \mu)(1 - \mathcal{R}_{0A}).$$

It can be seen that, $D_1 > 0$ whenever $\mathcal{R}_{0A} < 1$. Thus, the number of possible positive real roots the polynomial (10) can have depends on the signs of $D_2$, and $D_1$. Hence, the following results are established.

**Theorem 4**: The alcohol addiction sub-model (7)

A.  Has a unique alcohol addiction dominance equilibrium point if $\mathcal{R}_{0A} > 1$, and $D_2 < 0$.

B.  Could have a unique alcohol addiction dominance equilibrium point if $\mathcal{R}_{0A} > 1$, and $D_2 > 0$.

C.  Could have two alcohol addiction dominance equilibrium points if $\mathcal{R}_{0A} < 1$, and $D_2 < 0$.

The backward bifurcation in the sub-model (7) is shown in Part (C); that is, if a locally asymptotically stable alcohol addiction dominance equilibrium point and a locally asymptotically stable alcohol addiction free equilibrium point coexist whenever if $\mathcal{R}_{0A} < 1$; instances of this phenomenon's occurrence in mathematical models as well as its causes can be found in [38–40]. The physical result is that, while required, the traditional physical condition that the effective reproduction number be less than one is insufficient to completely effectively restrict the spread of alcohol addiction in the population. This section of the numerical simulation investigates the backward bifurcation phenomena in the sub-model (7).

## 3.2 Analysis of the smoking only sub-model

The smoking sub-model of the dynamical system (4) is determined by making $E_A = I_A = P_A = C_{SA} = P_{SA} = 0$, and it is given by

$$\frac{dS}{dt} = (1 - \pi)K - (\lambda_S + \delta + \mu)S,$$

$$\frac{dP}{dt} = \pi K + \delta S - (\delta_1\lambda_S + \mu)P,$$

$$\frac{dE_S}{dt} = \lambda_S S + \delta_1\lambda_S P - (\mu + \gamma_1)E_S, \tag{11}$$

$$\frac{dI_S}{dt} = \gamma_1 E_S - (\varepsilon_2 + \gamma_2 + \mu + d_S)I_S,$$

$$\frac{dP_S}{dt} = \gamma_2 I_S - (\mu + d_{PS})P_S,$$

$$\frac{dT}{dt} = \varepsilon_2 I_S - \mu T,$$

with smoking addiction state variables initial conditions $S(0){>}0$, $P(0){\geq}0$, $E_S(0){\geq}0$, $I_S(0){\geq}0$, $P_S(0){\geq}0$, $T(0){\geq}0$, total population $N_2(t) = S(t) + P(t) + E_S(t) + I_S(t) + P_S(t) + T(t)$, and smoking force of affection given by $\lambda_S = \frac{\beta_2}{N_2}(I_S(t) + P_S(t))$. Here like the smoking and alcohol dual addiction model, the smoking addiction sub-model is positively invariant in the region $\Omega_2 = \left\{ (S, P, E_S, I_S, P_S, T) \in \mathbb{R}^6_+, N_2 \leq \frac{\text{K}}{\mu} \right\}$, it is sufficient to consider the dynamics of model (11) in $\Omega_2$ be both physically and mathematically meaningful.

**3.2.1 Smoking addiction-free equilibrium point and its local stability.** Smoking addiction free equilibrium point of the smoking dissemination sub-model (11) is obtained by making its right-hand side as zero and setting the smoking addiction class and permanent smokers class to zero as $E_S = I_S = P_S = T = 0$. Then, we have computed the results as $S^0 = \frac{(1-\pi)\text{K}}{\delta+\mu}$, and $P^0 = \frac{\text{K}(\delta+\pi\mu)}{\mu(\delta+\mu)}$. Thus, the smoking dissemination sub-model (11) smoking free equilibrium point is given by $E_S^0 = \left( S^0, P^0, E_S^0, I_S^0, P_S^0, T^0 \right) = \left( \frac{(1-\pi)\text{K}}{\delta+\mu}, \frac{\text{K}(\delta+\pi\mu)}{\mu(\delta+\mu)}, 0, 0, 0, 0 \right)$.

Here, we are determining the smoking dissemination sub-model (11) effective reproduction number $\mathcal{R}_{0S}$ by utilizing the van den Driesch and Warmouth next-generation matrix technique [43]. Following lengthy calculations, we now have the dissemination matrix provided by

$$F = \begin{bmatrix} 0 & \frac{\beta_2}{N_2}S^0 + \frac{\beta_2}{N_2}\delta_1 P^0 & \frac{\beta_2}{N_2}S^0 + \frac{\beta_2}{N_2}\delta_1 P^0 & 0 \\ 0 & 0 & 0 & 0 \\ 0 & 0 & 0 & 0 \\ 0 & 0 & 0 & 0 \end{bmatrix},$$

and the transition matrix given by

$$V = \begin{bmatrix} (\mu + \gamma_1) & 0 & 0 & 0 \\ -\gamma_1 & (\varepsilon_2 + \gamma_2 + \mu + d_S) & 0 & 0 \\ 0 & -\gamma_2 & \mu + d_{PS} & 0 \\ 0 & -\varepsilon_2 & 0 & \mu \end{bmatrix}.$$

Then using Mathematica we have computed the result given by

$$V^{-1} = \begin{bmatrix} \dfrac{1}{(\varepsilon_2 + \gamma_2 + \mu)\mu} & 0 & 0 & 0 \\ \dfrac{\gamma_1}{(\mu + \gamma_1)(\varepsilon_2 + \gamma_2 + \mu)} & \dfrac{1}{(\varepsilon_2 + \gamma_2 + \mu + d_S)} & 0 & 0 \\ \dfrac{\gamma_2 \gamma_1}{(\mu + \gamma_1)(\varepsilon_2 + \gamma_2 + \mu)\mu} & \dfrac{\gamma_2}{(\varepsilon_2 + \gamma_2 + \mu)\mu} & \dfrac{1}{\mu + d_{PS}} & 0 \\ \dfrac{1}{\mu} & \dfrac{\varepsilon_2}{(\varepsilon_2 + \gamma_2 + \mu)\mu} & 0 & \dfrac{1}{\mu} \end{bmatrix},$$

and

$$FV^{-1} = \begin{bmatrix} \dfrac{\beta_2\gamma_1(\mu+\gamma_2)(S^0+\delta_1 P^0)}{N_2(\mu+\gamma_1)(\varepsilon_2+\gamma_2+\mu+d_S)\mu} & \dfrac{\beta_2(\mu+\gamma_2)(S^0+\delta_1 P^0)}{N_2(\varepsilon_2+\gamma_2+\mu+d_S)\mu} & \dfrac{\beta_2(S^0+\delta_1 P^0)}{N_2(\mu+d_{PS})} & 0 \\ 0 & 0 & 0 & 0 \\ 0 & 0 & 0 & 0 \\ 0 & 0 & 0 & 0 \end{bmatrix}.$$

The characteristic equation of the matrix $FV^{-1}$ is

$$\begin{vmatrix} \dfrac{\beta_2\gamma_1(\mu+\gamma_2)(S^0+\delta_1 P^0)}{N_2(\mu+\gamma_1)(\varepsilon_2+\gamma_2+\mu)\mu} - \lambda & \dfrac{\beta_2(\mu+\gamma_2)(S^0+\delta_1 P^0)}{N_2(\varepsilon_2+\gamma_2+\mu+d_S)\mu} & \dfrac{\beta_2(S^0+\delta_1 P^0)}{N_2(\mu+d_{PS})} & 0 \\ 0 & 0-\lambda & 0 & 0 \\ 0 & 0 & 0-\lambda & 0 \\ 0 & 0 & 0 & 0-\lambda \end{vmatrix} = 0.$$

Then the spectral radius (effective reproduction number $\mathcal{R}_{0S}$) of $FV^{-1}$ of the smoking addiction sub-model (11) is $\mathcal{R}_{0S} = \frac{\beta_2\gamma_1\mu(\mu+\gamma_2)(1-\pi)}{(\mu+\gamma_1)(\varepsilon_2+\gamma_2+\mu+d_S)(\mu++d_{PS})(\delta+\mu)} + \frac{\beta_2\gamma_1\delta_1(\mu+\gamma_2)(\pi\mu+\delta)}{(\mu+\gamma_1)(\varepsilon_2+\gamma_2+\mu+d_S)(\mu++d_{PS})(\delta+\mu)}.$

**Theorem 5**: The smoking addiction free equilibrium point $E_S^0$ of the smoking addiction dissemination sub-model (11) is locally asymptotically stable if $\mathcal{R}_{0S}<1$, otherwise unstable.

**Proof**: The local stability of the smoking addiction free equilibrium point of the system (11) at $E_S^0 = \left( \frac{(1-\pi)K}{\delta+\mu}, \frac{K(\delta+\pi\mu)}{\mu(\delta+\mu)}, 0, 0, 0, 0 \right)$ can be studied from its Jacobian matrix and Routh-Hurwitz stability criteria. The Jacobian matrix of the dynamical system at the smoking addiction free equilibrium point is given by

$$J(E_S^0) = \begin{bmatrix} -(\delta+\mu) & 0 & 0 & -\dfrac{\beta_2 S^0}{S^0+P^0} & -\dfrac{\beta_2 S^0}{S^0+P^0} & 0 \\ \delta & -\mu & 0 & -\dfrac{\beta_2\delta_1 P^0}{S^0+P^0} & -\dfrac{\beta_2\delta_1 P^0}{S^0+P^0} & 0 \\ 0 & 0 & -(\mu+\gamma_1) & \dfrac{\beta_2 S^0}{S^0+P^0}+\dfrac{\beta_2\delta_1 P^0}{S^0+P^0} & \dfrac{\beta_2 S^0}{S^0+P^0}+\dfrac{\beta_2\delta_1 P^0}{S^0+P^0} & 0 \\ 0 & 0 & \gamma_1 & -(\gamma_2+\gamma_2+\mu+d_S) & 0 & 0 \\ 0 & 0 & 0 & \gamma_2 & -\mu-d_{PS} & 0 \\ 0 & 0 & 0 & \varepsilon_2 & 0 & -\mu \end{bmatrix}.$$

Then the characteristic equation of the above Jacobian matrix is given by

$$
\begin{vmatrix}
-(\delta+\mu)-\lambda & 0 & 0 & -\dfrac{\beta_2 S^0}{S^0+P^0} & -\dfrac{\beta_2 S^0}{S^0+P^0} & 0 \\[2mm]
\delta & -\mu-\lambda & 0 & -\dfrac{\beta_2 \delta_1 P^0}{S^0+P^0} & -\dfrac{\beta_2 \delta_1 P^0}{S^0+P^0} & 0 \\[2mm]
0 & 0 & -(\mu+\gamma_1)-\lambda & \dfrac{\beta_2(S^0+\delta_1 P^0)}{S^0+P^0} & \dfrac{\beta_2(S^0+\delta_1 P^0)}{S^0+P^0} & 0 \\[2mm]
0 & 0 & \gamma_1 & -(\varepsilon_2+\gamma_2+\mu+d_S)-\lambda & 0 & 0 \\[2mm]
0 & 0 & 0 & \gamma_2 & -\mu-d_{PS}-\lambda & 0 \\[2mm]
0 & 0 & 0 & \varepsilon_2 & 0 & -\mu-\lambda
\end{vmatrix}
$$

$$= 0.$$

After some steps of calculations we have determined the results as $\lambda_1 = -\mu<0$ or $\lambda_2 = -(\gamma_2+\mu)<0$ or $\lambda^3 + [3\mu + \gamma_1 + \varepsilon_2 + \gamma_2]\lambda^2 + [[(\mu+\gamma_1)\mu + (2\mu+\gamma_1)(\varepsilon_2 + \gamma_2 + \mu + d_S)] - \frac{\beta_2\gamma_1(S^0+\delta_1 P^0)}{S^0+P^0}]\lambda + (\mu+\gamma_1)(\mu+d_{PS})(2\mu+\gamma_1)(\varepsilon_2 + \gamma_2 + \mu + d_S)[1-\mathcal{R}_{0S}] = 0$.

When $\mathcal{R}_{0S}<1$, all of the degree three polynomial equation eigenvalues are negative according to the Routh-Hurwiz stability requirements. As a result, the smoking addiction-free equilibrium point of the smoking addiction dissemination sub-model (11) is locally asymptotically stable, similar to [45], as all of the eigenvalues of the characteristics polynomials of the system (11) are negative if $\mathcal{R}_{0S}<1$.

**3.2.2 Smoking persistence equilibrium point (s).** Before checking the global stability of the smoking addiction-free equilibrium point of the smoking addiction dissemination sub-model, we shall find the possible number of smoking addiction persistence equilibrium point (s) of the model (11). Let $E_S^* = (S^*, P^*, E_S^*, I_S^*, P_S^*, T^*)$ be the smoking addiction dominance equilibrium point of the sub-model (11) and $\lambda_S^* = \frac{\beta_2}{N_2}\left(I_S^* + P_S^*\right)$ be the smoking incidence rate at the equilibrium point. To find persistence equilibrium point(s) for which smoking addiction is disseminating in the population, the Eq (11) are solved in terms of $\lambda_S^* = \frac{\beta_2}{N_2}\left(I_S^* + P_S^*\right)$ at smoking addiction persistence equilibrium point. Now setting the right-hand sides of the equations of the model (11) to zero (at steady state) gives

$$
S^* = \frac{(1-\pi)\mathrm{K}}{(\lambda_S^* + \delta + \mu)}, \quad P^* = \frac{\pi \mathrm{K}(\lambda_S^* + \delta + \mu) + (1-\pi)\mathrm{K}\delta}{(\delta_1 \lambda_S^* + \mu)(\lambda_S^* + \delta + \mu)},
$$

$$
E_S^* = \frac{\mathrm{K}\delta_1(\lambda_S^*)^2 + \pi \mathrm{K}\delta_1 \lambda_S^*(\delta + \mu) + (1-\pi)\mathrm{K}\lambda_S^*(\mu + \delta\delta_1)}{(\delta_1 \lambda_S^* + \mu)(\lambda_S^* + \delta + \mu)(\mu + \gamma_1)},
$$

$$
I_S^* = \frac{\gamma_1[\mathrm{K}\delta_1(\lambda_S^*)^2 + \pi \mathrm{K}\delta_1 \lambda_S^*(\delta + \mu) + (1-\pi)\mathrm{K}\lambda_S^*(\mu + \delta\delta_1)]}{(\delta_1 \lambda_S^* + \mu)(\lambda_S^* + \delta + \mu)(\varepsilon_2 + \gamma_2 + \mu + d_S)(\mu + \gamma_1)},
$$

$$
P_S^* = \frac{\gamma_1\gamma_2[\mathrm{K}\delta_1(\lambda_S^*)^2 + \pi \mathrm{K}\delta_1 \lambda_S^*(\delta + \mu) + (1-\pi)\mathrm{K}\lambda_S^*(\mu + \delta\delta_1)]}{(\delta_1 \lambda_S^* + \mu)(\lambda_S^* + \delta + \mu)(\varepsilon_2 + \gamma_2 + \mu + d_S)(\mu + d_{PS})(\mu + \gamma_1)},
$$

and

$$T^* = \frac{\gamma_1 \varepsilon_2 [K\delta_1 (\lambda_S^*)^2 + \pi K \delta_1 \lambda_S^* (\delta + \mu) + (1 - \pi) K \lambda_S^* (\mu + \delta \delta_1)]}{(\delta_1 \lambda_S^* + \mu)(\lambda_S^* + \delta + \mu)(\varepsilon_2 + \gamma_2 + \mu + d_S)(\mu + \gamma_1)(\mu + d_{PS})}.$$

Here we have substituted the expressions $S^*$, $P^*$, $E_S^*$, $I_S^*$, $P_S^*$, and $T^*$ in the smoking addiction incidence rate given by $\lambda_S^* = \frac{\beta_2}{N_2^*}\left(I_S^* + P_S^*\right)$, then we have computed and simplified the result to determine the second degree polynomial equation with $\lambda_S^*$ as

$$b_2(\lambda_S^*)^2 + b_1 \lambda_S^* + b_0 = 0, \tag{12}$$

where

$$b_2 = \frac{\mu(\varepsilon_2 + \gamma_2 + \mu)\mu + \gamma_1 \gamma_2 \mu + \gamma_1 \varepsilon_2 \mu + \gamma_1 \mu * \mu}{\mu(\varepsilon_2 + \gamma_2 + \mu + d_S)(\mu + d_{PS})} > 0,$$

$$b_1 = ((K\delta_1(\pi\mu + \delta) + K\mu(1 - \pi))b_2 + (\mu + \gamma_1)(\pi K + (1 - \pi)K\delta_1))(\varepsilon_2 + \gamma_2 + \mu + d_S)\mu - \beta_2 \gamma_1 (\mu + \gamma_2)K\delta_1,$$

$$b_0 = K(\varepsilon_2 + \gamma_2 + \mu + d_S)(\mu + d_{PS})(\mu + \gamma_1)(\delta + \mu)(1 - \mathcal{R}_{0S}) > 0 \text{ if } \mathcal{R}_{0S} < 1.$$

It can be seen from Eq (12) that, $b_2 > 0$ whenever $\mathcal{R}_{0S} < 1$. Thus, the number of possible positive real roots the polynomial (12) can have depends on the signs of $b_1$, and $b_0$. Hence, the following results are established.

**Theorem 6**: The smoking addiction sub-model model (11)

A. Has a unique smoking addiction persistence equilibrium point if $\mathcal{R}_{0S} > 1$, and $b_1 < 0$.

B. Has a unique smoking addiction persistence equilibrium point if $\mathcal{R}_{0S} > 1$, and $b_1 > 0$.

C. Could have two smoking addiction persistence equilibrium points if $\mathcal{R}_{0S} < 1$, and $b_1 < 0$.

A locally asymptotically stable smoking addiction persistence equilibrium point coexists with a locally asymptotically stable smoking addiction-free equilibrium point in the model (11) as revealed by Part (C). Examples of the occurrence of backward bifurcation phenomena in mathematical models, as well as their causes, can be found in [38, 39, 45]. The social implication is that, albeit being important, the traditional criterion that the smoking addiction's effective reproduction number be less than one is insufficient to effectively manage the addiction's spread across the society. The backward bifurcation curve in the numerical simulation section demonstrates the presence of the backward bifurcation phenomena in sub-model (11).

### 3.3 Alcohol and smoking dual addiction dissemination model analysis

Having analyzed the dynamics of the two sub-models, that is alcohol addiction sub- model (7) and the smoking addiction sub-model (11), the complete alcohol and smoking dual addiction dissemination model (4) is now considered (the analysis is done in the positively invariant region $\Omega$ given in Eq (6)).

**3.3.1 Dual addiction-free equilibrium point.** The alcohol and smoking dual addiction-free equilibrium point of the complete dynamical system (4) is obtained by taking the state variable $I_S = P_S = E_S = E_A = I_A = P_A = C_{SA} = P_{SA} = T = 0$ and is computed by $E_{0SA} = \left(\frac{(1-\pi)K}{\delta+\mu}, \frac{K(\pi\mu+\delta)}{\mu(\delta+\mu)}, 0, 0, 0, 0, 0, 0, 0, 0\right)$.

**3.3.2 Effective reproduction number of the dual addiction model.** The effective reproduction number of the dynamical system (4) is computed by applying the next generation operator method [43] and is the largest (dominant) eigenvalue (spectral radius) of the matrix:

$FV^{-1} = \left[ \frac{\partial \mathcal{F}_i(E_{0AS})}{\partial X_j} \right] \left[ \frac{\partial v_i(E_{0AS})}{\partial X_j} \right]$, where $\mathcal{F}_i$ is the rate of appearance of new entrance in compartment $i$, $v_i$ is the transfer of affected individuals from one compartment $i$ to another, and $E_{OSA}$ is the co-existence free equilibrium point. Here, we obtained the following matrices:

$$
\mathcal{F} = \begin{pmatrix}
\lambda_S S + \delta_1 \lambda_S P \\
\lambda_A S + \rho_1 \lambda_A P \\
0 \\
0 \\
0 \\
0 \\
0 \\
0 \\
0 \\
0 \\
0
\end{pmatrix},
$$

and

$$
\mathbb{V} = \begin{pmatrix}
(\mu + \gamma_1)E_S \\
(\mu + \alpha_1)E_A \\
(\varepsilon_2 + \gamma_2 + \rho_3 \lambda_A + \mu + d_S)I_S - \gamma_1 E_S \\
(\rho_2 \lambda_A + \mu + d_{PS})P_S - \gamma_2 I_S \\
(\alpha_2 + \varepsilon_1 + \delta_2 \lambda_S + \mu + d_A)I_A - \alpha_1 E_A \\
(\delta_3 \lambda_S + \mu + d_{PA})P_A - \alpha_2 I_A \\
(\alpha + \varepsilon_3 + \mu + d_C)C_{SA} - \delta_2 \lambda_S I_A - \delta_3 \lambda_S P_A - \rho_3 \lambda_A I_S - \rho_2 \lambda_A P_S \\
(\mu + d_{SA})P_{SA} - \alpha C_{SA} \\
\mu T - \varepsilon_3 C_{SA} - \varepsilon_1 I_A - \varepsilon_2 I_S \\
(\rho_1 \lambda_A + \delta_1 \lambda_S + \mu)P - \pi K - \delta S \\
(\lambda_S + \lambda_A + \delta + \mu)S - (1 - \pi)K
\end{pmatrix}.
$$

Then, the spectral radius of the corresponding next generation matrix, $FV^{-1}$ is given in terms of smoking and alcohol dual addiction associated effective reproduction number. It is therefore evident that the associated effective reproduction number for the smoking and alcohol dual addiction dissemination model denoted by $R_{0AS}$ is given by $\mathcal{R}_{0AS} = max\{\mathcal{R}_{0A}, \mathcal{R}_{0S}\}$, where $\mathcal{R}_{0A} = \frac{\beta_1 \alpha_1 \mu (\mu + \alpha_2)(1 - \pi)}{(\mu + \alpha_1)(\alpha_2 + \varepsilon_1 + \mu + d_A)(\mu + d_{PA})(\delta + \mu)} + \frac{\beta_1 \alpha_1 \rho_1 (\mu + \alpha_2)(\delta + \pi \mu)}{(\mu + \alpha_1)(\alpha_2 + \varepsilon_1 + \mu + d_A)(\mu + d_{PA})(\delta + \mu)}$, $\mathcal{R}_{0S} = \frac{\beta_2 \gamma_1 \mu (\mu + \delta)(1 - \pi)}{(\mu + \gamma_1)(\varepsilon_2 + \gamma_2 + \mu + d_S)(\mu + d_{PS})(\delta + \mu)} + \frac{\beta_2 \gamma_1 \delta_1 (\mu + \gamma_2)(\varepsilon \mu + \delta)}{(\mu + \sigma)(\varepsilon_2 + \gamma_2 + \mu + d_S)(\mu + d_{PS})(\delta + \mu)}$.

Here $\mathcal{R}_{0AS}$ is defined as the average number of secondary smoking and alcohol dual addicted individuals produced by one dual addicted individual who live in a whole individuals who are at risk for dual addiction during his/her addiction period. The threshold result $\mathcal{R}_{0AS}$ is the effective reproduction number for the smoking and alcohol dual addicted individuals. Based on the dual addiction effective reproduction number $\mathcal{R}_{0AS} = max\{\mathcal{R}_{0A}, \mathcal{R}_{0S}\}$, where the alcohol only addiction dissemination sub-model (7) effective reproduction number is represented by

$\mathcal{R}_{0A} = \frac{\beta_1 \alpha_1 \mu (\mu+\alpha_2)(1-\pi)}{(\mu+\alpha_1)(\alpha_2+\varepsilon_1+\mu+d_A)(\mu+d_{PA})(\delta+\mu)} + \frac{\beta_1 \alpha_1 \rho_1 (\mu+\alpha_2)(\delta+\pi\mu)}{(\mu+\alpha_1)(\alpha_2+\varepsilon_1+\mu+d_A)(\mu+d_{PA})(\delta+\mu)}$, and the smoking only addiction dissemination sub-model (11) effective reproduction number is represented by

$\mathcal{R}_{0S} = \frac{\beta_2 \gamma_1 \mu (\mu+\delta)(1-\pi)}{(\mu+\gamma_1)(\varepsilon_2+\gamma_2+\mu+d_S)(\mu+d_{PS})(\delta+\mu)} + \frac{\beta_2 \gamma_1 \delta_1 (\mu+\gamma_2)(\varepsilon\mu+\delta)}{(\mu+\sigma)(\varepsilon_2+\gamma_2+\mu+d_S)(\mu+d_{PS})(\delta+\mu)}$.

Whenever $\mathcal{R}_{0AS}<1$, the dual addiction dissemination model exhibit the phenomenon of backward bifurcation, i.e. the dual addiction may not be eliminated in the near future without implementing further control efforts, but if $\mathcal{R}_{0AS}>1$, the dual addiction will disseminate in the community. In order to minimize the dual addiction model effective reproduction number $\mathcal{R}_{0AS}$, we can vary the model parameters incorporated in $\mathcal{R}_{0AS}$. Since $\mathcal{R}_{0AS}$ is dependent on the model parameters $\beta_1, \alpha_1, \mu, \alpha_2, \pi, \varepsilon_1, d_A, d_{PA}, \delta, \rho_1, \beta_2, \gamma_1, \varepsilon_2, \gamma_2, d_S, \delta, \delta_1$, and $d_{PS}$. From the sensitivity indices results illustrated in Table 3 and the sensitivity indices diagram given by Fig 2 the dual addiction model effective reproduction number is directly proportional to some of the model parameters such as $\beta_1, \alpha_1, \mu, \alpha_2, \rho_1, \beta_2, \gamma_1, \varepsilon_2, \gamma_2$, and $\delta_1$ and also inversely proportional to some of the model parameters like $\pi, \varepsilon_1, d_A, d_{PA}, \delta, \varepsilon_2, d_S$, and $d_{PS}$.

**3.3.3 Locally asymptotic stability of the dual addiction-free equilibrium.** **Theorem 7:** The smoking and alcohol dual addiction-free equilibrium point $E_{0AS}$ of the complete dynamical system (4) is locally asymptotically stable whenever $\mathcal{R}_{0AS}<1$, otherwise unstable.

**Proof:** The local stability of the smoking and alcohol dual addiction free equilibrium point of the complete dynamical system (4) can be studied from its Jacobian matrix at the smoking and alcohol dual addiction-free equilibrium point $E_{0AS}$ and Routh-Hurwitz stability criteria [41]. Then, the Jacobian matrix of the dynamical system (4) at $E_{0AS} =$

$\left( \frac{(1-\pi)K}{\delta+\mu}, \frac{K(\pi\mu+\delta)}{\mu(\delta+\mu)}, 0, 0, 0, 0, 0, 0, 0, 0, 0 \right)$ is given by

$$
J(E_{0AS}) = \begin{pmatrix}
-A_4 & 0 & 0 & -\frac{\beta_2}{N^0}S^0 & -\frac{\beta_2}{N^0}S^0 & 0 & -\frac{\beta_1}{N^0}S^0 & -\frac{\beta_1}{N^0}S^0 & -A_0 & -A_0 & 0 \\
\delta & -\mu & 0 & -\frac{\beta_2}{N^0}P^0 & -\frac{\beta_2}{N^0}P^0 & 0 & -\frac{\beta_1}{N^0}P^0 & -\frac{\beta_1}{N^0}P^0 & -A_1 & -A_1 & 0 \\
0 & 0 & -A_5 & A_2 & A_2 & 0 & 0 & 0 & \theta_1 A_2 & \theta_1 A_2 & 0 \\
0 & 0 & \gamma_1 & -A_6 & 0 & 0 & 0 & 0 & 0 & 0 & 0 \\
0 & 0 & 0 & \delta & -A_7 & 0 & 0 & 0 & 0 & 0 & 0 \\
0 & 0 & 0 & A_3 & A_3 & -A_8 & 0 & 0 & \alpha_1 A_3 & \varphi A_3 & 0 \\
0 & 0 & 0 & 0 & 0 & \alpha_1 & -A_9 & 0 & 0 & 0 & 0 \\
0 & 0 & 0 & 0 & 0 & 0 & \alpha_2 & -A_{10} & 0 & 0 & 0 \\
0 & 0 & 0 & 0 & 0 & 0 & 0 & 0 & -A_{11} & 0 & 0 \\
0 & 0 & 0 & 0 & 0 & 0 & 0 & 0 & \alpha & -A_{12} & 0 \\
0 & 0 & 0 & \varepsilon_2 & 0 & 0 & \varepsilon_1 & 0 & \varepsilon_3 & 0 & -\mu
\end{pmatrix},
$$

**Table 3. Sensitivity indices whenever $\mathcal{R}_{0AS} = \max\{\mathcal{R}_{0A}, \mathcal{R}_{0S}\} = \mathcal{R}_{0A}$.**

| Parameters | Sensitivity indices |
|---|---|
| $\beta_1$ | $SI_{\beta_1}^{\mathcal{R}_{0A}} = +1$ |
| $\alpha_1$ | $SI_{\alpha_1}^{\mathcal{R}_{0A}} = +0.82$ |
| $\alpha_2$ | $SI_{\alpha_2}^{\mathcal{R}_{0A}} = +0.41$ |
| $\pi$ | $SI_{\pi}^{\mathcal{R}_{0A}} = -0.87$ |
| $\rho_1$ | $SI_{\rho_1}^{\mathcal{R}_{0A}} = +0.53$ |
| $\delta$ | $SI_{\delta}^{\mathcal{R}_{0A}} = -0.84$ |
| $\varepsilon_1$ | $SI_{\varepsilon_1}^{\mathcal{R}_{0A}} = -0.76$ |
| $d_A$ | $SI_{d_A}^{\mathcal{R}_{0A}} = -0.43$ |
| $d_{PA}$ | $SI_{d_{PA}}^{\mathcal{R}_{0A}} = -0.38$ |

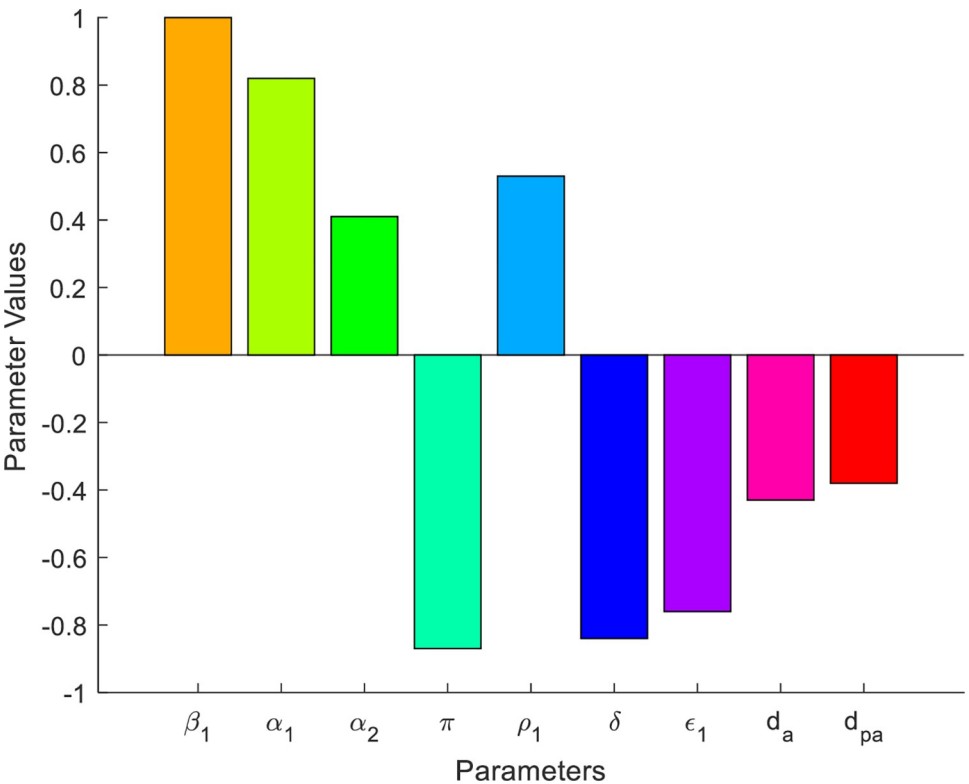

**Fig 2. Model parameters sensitivity indices whenever max$\{\mathcal{R}_{0A}, \mathcal{R}_{0S}\} = \mathcal{R}_{0A} = 1.74 > 1$.**

where, $A_0 = \left(\alpha_1 \frac{\beta_1}{N^0} + \theta_1 \frac{\beta_2}{N^0}\right) S^0$, $A_1 = \left(\alpha_1 \frac{\beta_1}{N^0} + \theta_1 \frac{\beta_2}{N^0}\right) P^0$, $A_2 = \frac{\beta_2}{N^0}\left(S^0 + \delta_1 P^0\right)$, $A_3 = \frac{\beta_1}{N^0}\left(S^0 + \rho_1 P^0\right)$, $A_4 = (\delta + \mu)$, $A_5 = (\mu + \gamma_1)$, $A_6 = (\varepsilon_2 + \gamma_2 + \mu + d_S)$, $A_7 = \mu$, $A_8 = (\mu + \alpha_1)$, $A_9 = (\alpha_2 + \varepsilon_1 + \mu + d_A)$, $A_{10} = \mu$, $A_{11} = (\alpha + \varepsilon_3 + \mu + d_C)$ and $A_{12} = \mu$.

Then, the characteristic equation of the above Jacobian matrix is given by

$$
\begin{vmatrix}
-A_4 - \lambda & 0 & 0 & -\frac{\beta_2}{N^0}S^0 & -\frac{\beta_2}{N^0}S^0 & 0 & -\frac{\beta_1}{N^0}S^0 & -\frac{\beta_1}{N^0}S^0 & -A_0 & -A_0 & 0 \\
\delta & -\mu - \lambda & 0 & -\frac{\beta_2}{N^0}P^0 & -\frac{\beta_2}{N^0}S^0 & 0 & -\frac{\beta_1}{N^0}P^0 & -\frac{\beta_1}{N^0}P^0 & -A_1 & -A_1 & 0 \\
0 & 0 & -A_5 - \lambda & A_2 & A_2 & 0 & 0 & 0 & \theta_1 A_2 & \theta_1 A_2 & 0 \\
0 & 0 & \gamma_1 & -A_6 - \lambda & 0 & 0 & 0 & 0 & 0 & 0 & 0 \\
0 & 0 & 0 & \gamma_2 & -A_7 - \lambda & 0 & 0 & 0 & 0 & 0 & 0 \\
0 & 0 & 0 & A_3 & A_3 & -A_8 - \lambda & 0 & 0 & \alpha_1 A_3 & \alpha_1 A_3 & 0 \\
0 & 0 & 0 & 0 & 0 & \alpha_1 & -A_9 - \lambda & 0 & 0 & 0 & 0 \\
0 & 0 & 0 & 0 & 0 & 0 & \alpha_2 & -A_{10} - \lambda & 0 & 0 & 0 \\
0 & 0 & 0 & 0 & 0 & 0 & 0 & 0 & -A_{11} - \lambda & 0 & 0 \\
0 & 0 & 0 & 0 & 0 & 0 & 0 & 0 & \alpha & -A_{11} - \lambda & 0 \\
0 & 0 & 0 & \varepsilon_2 & 0 & 0 & \varepsilon_1 & 0 & \varepsilon_3 & 0 & -\mu - \lambda
\end{vmatrix}
$$

$= 0$.

After some steps of computations we have got $\lambda = -\mu, \lambda = -\mu < 0, \lambda = -A_4 < 0, \lambda = -A_{10} < 0, \lambda = -A9 < 0, \lambda = -A_8 < 0, \lambda = -A_{11} < 0\mu, \lambda = -A_{12} < 0$, or

$$\lambda^3 + [A_7 + A_6 + A_5]\lambda^2 + [A_7(A_6 + A_5) + A_6 A_5 - \gamma_1 A_2]\lambda + \mu(\varepsilon_2 + \gamma_2 + \mu)(\mu + \gamma_1)(1 - \mathcal{R}_{0A}) = 0. \quad (13)$$

Now by applying the Routh-Hurwiz stability criteria stated in [41], we can reveal that each of the eigenvalue of the polynomial is negative. Therefore, since all the eigenvalues of the characteristics polynomials of the system (4) are negative whenever $\mathcal{R}_{0AS} < 1$, the smoking and alcohol dual addiction-free equilibrium point of the smoking and alcohol dual addiction dissemination model (4) is locally asymptotically stable.

**3.3.4 Smoking and alcohol dual addiction persistence equilibrium point.** Persistence equilibrium points are steady-state solutions where both smoking addiction and alcohol addiction persists in the population (all state variables are positive). Let the smoking and alcohol dual addiction persistence equilibrium of the complete model system (4) is denoted by $E_{AS}^* = (S^*, P^*, E_S^*, I_S^*, P_S^*, E_A^*, I_A^*, P_A^*, C_{SA}^*, P_{SA}^*, T^*)$. It is obtained by setting the right hand side of each equation of the dual addiction dissemination model (4) equal to zero, and computed as

$$S^* = \frac{(1-\pi)K}{(\lambda_S^* + \lambda_A^* + \delta + \mu)}, P^* = \frac{\pi K + \delta S^*}{(\rho_1 \lambda_A^* + \delta_1 \lambda_S^* + \mu)}, E_S^* = \frac{\lambda_S^* S^* + \delta_1 \lambda_S^* P^*}{(\mu + \gamma_1)}, I_S^*$$

$$= \frac{\gamma_1 E_S^*}{(\varepsilon_2 + \gamma_2 + \rho_3 \lambda_A^* + \mu + d_S)}, P_S^* = \frac{\gamma_2 I_S^*}{(\rho_2 \lambda_A^* + \mu + d_{PS})}, E_A^* = \frac{\lambda_A^* S^* + \rho_1 \lambda_A^* P^*}{(\mu + \alpha_1)}, I_A^*$$

$$= \frac{\alpha_1 E_A^*}{(\alpha_2 + \varepsilon_1 + \delta_2 \lambda_S^* + \mu + d_A)}, P_A^* = \frac{\alpha_2 I_A^*}{(\delta_3 \lambda_S^* + \mu + d_{PA})},$$

$$C_{SA}^* = \frac{\delta_2 \lambda_S^* I_A^* + \delta_3 \lambda_S^* P_A^* + \rho_3 \lambda_A^* I_S^* + \rho_4 \lambda_A^* P_S^*}{(\alpha + \varepsilon_3 + \mu + d_C)}, P_{SA}^* = \frac{\alpha C_{SA}^*}{(\mu + d_{SA})}, T^* = \frac{\varepsilon_3 C_{SA}^* + \varepsilon_1 I_A^* + \varepsilon_2 I_S^*}{\mu}.$$

**3.3.5 Global dynamics of the dual addiction model at $\mathcal{R}_{0AS} = 1$.** **Theorem 8:** The smoking and alcoholism dual addiction (4) exhibits the phenomenon of backward bifurcation at $\mathcal{R}_{0AS} = 1$ whenever the inequality $K_1 > K_2$ holds.

**Proof:** Let us use the Centre Manifold criteria and change of dual addiction model state variables by $S = z_1, P = z_2, E_S = z_3, I_S = z_4, P_S = z_5, E_A = z_6, I_A = z_7, P_A = z_8, C_{SA} = z_9, P_{SA} = z_{10}$ and $T = z_{11}$.

Thus, $N(t) = z_1(t) + z_2(t) + z_3(t) + z_4(t) + z_5(t) + z_6(t) + z_7(t) + z_8(t) + z_9(t) + z_{10}(t) + z_{11}(t)$.

Moreover, based on the vector notation represented by

$Z = (z_1, z_2, z_3, z_4, z_5, z_6, z_7, z_8, z_9, z_{10}, z_{11})^T$, the smoking and alcoholism dual-addiction model (4) can be re-written in the form $\frac{dZ}{dt} = h(Z)$ where $H = (h_1, h_2, h_3, h_4, h_5, h_6, h_7, h_8, h_9, h_{10}, h_{11})^T$, such that

$$\frac{dz_1}{dt} = h_1 = (1-\pi)K - (\lambda_S + \lambda_A + \delta + \mu)z_1,$$

$$\frac{dz_2}{dt} = h_2 = \pi K + \delta z_1 - (\rho_1 \lambda_A + \delta_1 \lambda_S + \mu)z_2$$

$$\frac{dz_3}{dt} = h_3 = \lambda_S z_1 + \delta_1 \lambda_S z_2 - (\mu + \gamma_1)z_3, \quad (14)$$

$$\frac{dz_4}{dt} = h_4 = \gamma_1 z_3 - (\varepsilon_2 + \gamma_2 + \rho_3 \lambda_A + \mu + d_S)z_4,$$

$$\frac{dz_5}{dt} = h_5 = \gamma_2 z_4 - (\rho_2 \lambda_A + \mu + d_{PS}) z_5,$$

$$\frac{dz_6}{dt} = h_6 = \lambda_A z_1 + \rho_1 \lambda_A z_2 - (\mu + \alpha_1) z_6,$$

$$\frac{dz_7}{dt} = h_7 = \alpha_1 z_6 - (\alpha_2 + \varepsilon_1 + \delta_2 \lambda_S + \mu + d_A) z_7,$$

$$\frac{dz_8}{dt} = h_8 = \alpha_2 z_7 - (\delta_3 \lambda_S + \mu + d_{PA}) z_8,$$

$$\frac{dz_9}{dt} = h_9 = \delta_2 \lambda_S z_7 + \delta_3 \lambda_S z_8 + \rho_3 \lambda_A z_4 + \rho_2 \lambda_A z_5 - (\alpha + \varepsilon_3 + \mu + d_C) z_9,$$

$$\frac{dz_{10}}{dt} = h_{10} = \alpha z_9 - (\mu + d_{SA}) z_{10},$$

$$\frac{dz_{11}}{dt} = h_{11} = \varepsilon_3 z_9 + \varepsilon_1 z_7 + \varepsilon_2 z_4 - \mu z_{11},$$

with initial conditions represented by $z_1(0) = z_1^0 \geq 0, z_2(0) = z_2^0 \geq 0, z_3(0) = z_3^0 \geq 0,$ $z_4(0) = z_4^0 \geq 0, z_5(0) = z_5^0 \geq 0, z_6(0) = z_7^0 \geq 0, z_8(0) = z_8^0 \geq 0, z_9(0) = z_9^0 \geq 0,$ $z_{10}(0) = z_{10}^0 \geq 0,$ and $z_{11}(0) = z_{11}^0 \geq 0, \lambda_A(t) = \frac{\beta_1}{N(t)} (z_7(t) + z_8(t) + \theta_1(z_9(t) + z_{10}(t))),$ and $\lambda_S(t) = \frac{\beta_2}{N(t)} (z_4(t) + z_5(t) + \theta_2(z_9(t) + z_{10}(t)))$ where $\theta_1 \geq 1$ and $\theta_2 \geq 1$.

The Jacobian matrix of the Eq (14) at the dual addiction-free equilibrium point $E_{0AS}$ is the same as that of Eq (13). Now without loss of generality, let us consider the case when $\mathcal{R}_{0S} > \mathcal{R}_{0A}$ and $\mathcal{R}_{0AS} = 1$, so that $\mathcal{R}_{0S} = 1$. Moreover, let $\beta_2 = \beta^*$ is chosen as a parameter that will cause bifurcation phenomenon, then solving for $\beta_2$ from $\mathcal{R}_{0S} = 1$ we have determined that $\beta^* = \beta_2 = \frac{\mu(\mu + \gamma_1)(\varepsilon_2 + \gamma_2 + \mu + d_S)(\delta + \mu)(\mu + \sigma)}{\gamma_1[(\mu + d_{PS})(\mu + \delta)(1 - \pi)(\mu + \sigma) + \delta_1(\mu + \gamma_2)(\varepsilon\mu + \delta)]}$.

Now at $\beta^* = \beta_2$, the characteristics equation illustrated in Eq (13) has a single zero eigenvalue because $\mu(\varepsilon_2 + \gamma_2 + \mu)(\mu + \gamma_1)(1 - \mathcal{R}_{0A}) = 0$ and each of the remaining eigenvalues has negative real part. Now let us apply the Center Manifold Criteria on the new dynamical system (14) near to $\beta^* = \beta_2$. Then, the right and left eigenvectors $u = (u_1, u_2, u_3, u_4, u_5, u_6, u_7, u_8, u_9, u_{10}, u_{11})^T$ and $v = (v_1, v_2, v_3, v_4, v_5, v_6, v_7, v_8, v_9, v_{10}, v_{11})^T$ with respect to the zero eigenvalue of the Jacobian matrix $J(E_{0AS})$ evaluated at $\beta^* = \beta$ are respectively computed and represented by

$$u_1 = -\left( \frac{\beta_2 z_1^0 \gamma_1}{A_4 A_6 N^0} - \frac{\beta_2 z_1^0 \delta \gamma_1}{A_4 A_6 A_7 N^0} + \frac{\beta_1 \alpha_1}{A_4 A_9 N^0} \left( \frac{A_3 \gamma_1}{A_6} + \frac{\delta A_3 \gamma_1}{A_6 A_7} \right) + \frac{\beta_1 z_1^0 \alpha_1 \alpha_2}{A_4 A_9 A_{10} N^0} \left( \frac{A_3 \gamma_1}{A_6} + \frac{\delta A_3 \gamma_1}{A_6 A_7} \right) \right) u_3$$
$$< 0,$$

$$u_2 = \left[\frac{\beta_2 z_1^0 \delta\gamma_1}{A_4 A_6 \mu N^0} + \frac{\beta_2 z_1^0 \delta\delta\gamma_1}{A_4 A_6 A_7 \mu N^0} + \frac{\beta_1 \delta\alpha_1}{A_4 A_9 \mu N^0}\left(\frac{A_3\gamma_1}{A_6} + \frac{\delta A_3\gamma_1}{A_6 A_7}\right) + \frac{\beta_1 z_1^0 \delta\alpha_1\alpha_2}{A_4 A_9 A_{10} \mu N^0}\left(\frac{A_3\gamma_1}{A_6} + \frac{\delta A_3\gamma_1}{A_6 A_7}\right)\right)$$

$$- \frac{\beta_2}{\mu N^0} z_2^0 \frac{\gamma_1}{A_6} - \frac{\beta_2}{\mu N^0} z_2^0 \frac{\delta\gamma_1}{A_6 A_7} - \frac{\beta_1}{\mu N^0} z_2^0 \frac{\alpha_1}{A_9}\left(\frac{A_3\gamma_1}{A_6} + \frac{\delta A_3\gamma_1}{A_6 A_7}\right)$$

$$- \frac{\beta_1}{\mu N^0} z_2^0 \frac{\alpha_1\alpha_2}{A_9 A_{10}}\left(\frac{A_3\gamma_1}{A_6} + \frac{\delta A_3\gamma_1}{A_6 A_7}\right)\right]u_3, u_3$$

$$= u_3 > 0, u_4 = \frac{\gamma_1}{A_6} u_3 > 0, u_5 = \frac{\delta\gamma_1}{A_6 A_7} u_3 > 0, u_6 = \left(\frac{A_3\gamma_1}{A_6} + \frac{\delta A_3\gamma_1}{A_6 A_7}\right)u_3 > 0, u_7$$

$$= \frac{\alpha_1}{A_9}\left(\frac{A_3\gamma_1}{A_6} + \frac{\delta A_3\gamma_1}{A_6 A_7}\right)u_3 > 0, u_8 = \frac{\alpha_1\alpha_2}{A_9 A_{10}}\left(\frac{A_3\gamma_1}{A_6} + \frac{\delta A_3\gamma_1}{A_6 A_7}\right)u_3 > 0, u_9 = u_{10} = 0, u_{11}$$

$$= \left(\frac{\gamma_1\varepsilon_2}{A_6} + \frac{\alpha_1\varepsilon_1}{A_9}\left(\frac{A_3\gamma_1}{A_6} + \frac{\delta A_3\gamma_1}{A_6 A_7}\right)\right)u_3 > 0,$$

and

$$v_1 = v_2 = v_6 = v_7 = v_8 = v_{11} = 0, v_3 = v_3 > 0, v_4 = \frac{A_5}{\gamma_1}v_3 > 0, v_5 = \left(\frac{A_5 A_6}{\gamma_1} - A_2\right)v_3,$$

$$v_9 = \left(\frac{\theta_1 A_2}{A_{11}} + \frac{\alpha}{A_{11}}\frac{\theta_1 A_2}{A_{12}}\right)v_3 > 0, v_{10} = \frac{\theta_1 A_2}{A_{12}}v_3 > 0.$$

We found the non-zero partial derivatives with respect to the bifurcation parameter and the non-zero second-order mixed derivatives of h with respect to variables for the dynamical system (14) at the dual addiction-free equilibrium point $E_{0AS}$ in order to derive the bifurcation coefficients a and b. The following formulae are used to compute the coefficients:

$$a = \sum_{k,i,j=1}^{11} v_k u_i u_j \frac{\partial^2 h_k}{\partial z_i \partial z_j}(E_{0AS}, \beta^*), \text{ and } b = \sum_{k,i=1}^{11} v_k u_i \frac{\partial^2 h_k}{\partial z_i \partial \beta^*}(E_{0AS}, \beta^*).$$

The only non-zero second order partial derivatives at $E_{0AS}$ are given by

$$\frac{\partial^2 h_3}{\partial z_1 \partial z_4} = \frac{\partial^2 h_3}{\partial z_4 \partial z_1} = \frac{\beta_2}{z_1^0 + z_2^0}, \frac{\partial^2 h_3}{\partial z_2 \partial z_4} = \frac{\partial^2 h_3}{\partial z_4 \partial z_2} = \frac{\beta_2\delta_1}{z_1^0 + z_2^0}, \text{ and } \frac{\partial^2 h_3}{\partial \beta^* \partial z_4} = \frac{\partial^2 h_3}{\partial z_4 \partial \beta^*} = \frac{z_1^0}{z_1^0 + z_2^0} + \frac{\delta_1 z_2^0}{z_1^0 + z_2^0}.$$

Thus,

$$a = \sum_{i,j=1}^{11} v_3 u_i u_j \frac{\partial^2 h_3}{\partial z_i \partial z_j}(E_{0AS}, \beta^*) = 2v_3 u_1 u_4 \frac{\partial^2 h_3}{\partial z_1 \partial z_4} + 2v_3 u_2 u_4 \frac{\partial^2 h_3}{\partial z_2 \partial z_4},$$

$$= 2v_3\left[u_1 u_4 \frac{\beta_2}{z_1^0 + z_2^0} + u_2 u_4 \frac{\beta_2\delta_1}{z_1^0 + z_2^0}\right],$$

$$= 2v_3 u_3^2\left[-\frac{\beta_2\gamma_1}{A_6(z_1^0 + z_2^0)}\left(\frac{\beta_2 z_1^0\gamma_1}{A_4 A_6 N^0} + \frac{\beta_2 z_1^0 \delta\gamma_1}{A_4 A_6 A_7 N^0} + \frac{\beta_1\alpha_1}{A_4 A_9 N^0}\left(\frac{A_3\gamma_1}{A_6} + \frac{\delta A_3\gamma_1}{A_6 A_7}\right) + \frac{\beta_1 z_1^0 \alpha_1\alpha_2}{A_4 A_9 A_{10} N^0}\left(\frac{A_3\gamma_1}{A_6} + \frac{\delta A_3\gamma_1}{A_6 A_7}\right)\right)\right.$$

$$+ \frac{\beta_2\delta_1\gamma_1}{A_6(z_1^0 + z_2^0)}\left[\frac{\beta_2 z_1^0 \delta\gamma_1}{A_4 A_6 \mu N^0} + \frac{\beta_2 z_1^0 \delta\delta\gamma_1}{A_4 A_6 A_7 \mu N^0} + \frac{\beta_1 \delta\alpha_1}{A_4 A_9 \mu N^0}\left(\frac{A_3\gamma_1}{A_6} + \frac{\delta A_3\gamma_1}{A_6 A_7}\right) + \frac{\beta_1 z_1^0 \delta\alpha_1\alpha_2}{A_4 A_9 A_{10} \mu N^0}\left(\frac{A_3\gamma_1}{A_6} + \frac{\delta A_3\gamma_1}{A_6 A_7}\right)\right)$$

$$\left.- \left(\frac{\beta_2}{\mu N^0} z_2^0 \frac{\gamma_1}{A_6} + \frac{\beta_2}{\mu N^0} z_2^0 \frac{\delta\gamma_1}{A_6 A_7} + \frac{\beta_1}{\mu N^0} z_2^0 \frac{\alpha_1}{A_9}\left(\frac{A_3\gamma_1}{A_6} + \frac{\delta A_3\gamma_1}{A_6 A_7}\right) + \frac{\beta_1}{\mu N^0} z_2^0 \frac{\alpha_1\alpha_2}{A_9 A_{10}}\left(\frac{A_3\gamma_1}{A_6} + \frac{\delta A_3\gamma_1}{A_6 A_7}\right)\right)\right]\right],$$

$$= 2v_3 u_3^2 [K_1 - K_2] \text{ where}$$

$$K_1 = \frac{\beta_2 \delta_1 \gamma_1}{A_6(z_1^0 + z_2^0)} \left[ \frac{\beta_2 z_1^0 \delta \gamma_1}{A_4 A_6 \mu N^0} + \frac{\beta_2 z_1^0 \delta \delta \gamma_1}{A_4 A_6 A_7 \mu N^0} + \frac{\beta_1 \delta \alpha_1}{A_4 A_9 \mu N^0} \left( \frac{A_3 \gamma_1}{A_6} + \frac{\delta A_3 \gamma_1}{A_6 A_7} \right) + \frac{\beta_1 z_1^0 \delta \alpha_1 \alpha_2}{A_4 A_9 A_{10} \mu N^0} \left( \frac{A_3 \gamma_1}{A_6} + \frac{\delta A_3 \gamma_1}{A_6 A_7} \right) \right) ],$$

$$K_2 = \frac{\beta_2 \gamma_1}{A_6(z_1^0 + z_2^0)} \left( \frac{\beta_2 z_1^0 \gamma_1}{A_4 A_6 N^0} + \frac{\beta_2 z_1^0 \delta \gamma_1}{A_4 A_6 A_7 N^0} + \frac{\beta_1 \alpha_1}{A_4 A_9 N^0} \left( \frac{A_3 \gamma_1}{A_6} + \frac{\delta A_3 \gamma_1}{A_6 A_7} \right) + \frac{\beta_1 z_1^0 \alpha_1 \alpha_2}{A_4 A_9 A_{10} N^0} \left( \frac{A_3 \gamma_1}{A_6} + \frac{\delta A_3 \gamma_1}{A_6 A_7} \right) \right)$$
$$+ \left( \frac{\beta_2}{\mu N^0} z_2^0 \frac{\gamma_1}{A_6} + \frac{\beta_2}{\mu N^0} z_2^0 \frac{\delta \gamma_1}{A_6 A_7} + \frac{\beta_1}{\mu N^0} z_2^0 \frac{\alpha_1}{A_9} \left( \frac{A_3 \gamma_1}{A_6} + \frac{\delta A_3 \gamma_1}{A_6 A_7} \right) + \frac{\beta_1}{\mu N^0} z_2^0 \frac{\alpha_1 \alpha_2}{A_9 A_{10}} \left( \frac{A_3 \gamma_1}{A_6} + \frac{\delta A_3 \gamma_1}{A_6 A_7} \right) \right).$$

Similarly, we compute the result represented by

$$b = \sum_{k,i=1}^{11} v_k u_i \frac{\partial^2 h_k}{\partial z_i \partial \beta^*} (E_{0AS}, \beta^*) = \frac{\partial^2 h_3}{\partial \beta^* \partial z_4} + \frac{\partial^2 h_3}{\partial z_4 \partial \beta^*} = \frac{2(z_1^0 + \delta_1 z_2^0)}{z_1^0 + z_2^0} > 0.$$

Finally, since the bifurcation coefficients $a = 2v_3 u_3^2 [K_1 - K_2] > 0$ whenever $K_1 > K_2$ and $b > 0$ the smoking and alcoholism dual addiction free equilibrium point $E_{0AS}$ of the model (4) is not globally asymptotically stable at $\mathcal{R}_{0AS} = \max\{\mathcal{R}_{0A}, \mathcal{R}_{0S}\} = 1$. Thus, the dual addiction model (4) shows bifurcation in the backward direction whenever $\mathcal{R}_{0AS} = \max\{\mathcal{R}_{0A}, \mathcal{R}_{0S}\} = 1$.

**3.3.6 Sensitivity analysis.** We conducted a sensitivity analysis of the smoking and alcohol dual addiction model parameters included in the effective reproduction numbers in this subsection since our study takes optimal control theory into consideration. It is vital to look at the most important model parameters that might change the threshold quantity (also known as the dual addiction dissemination model effective reproduction number, or $\mathcal{R}_{0AS}$) between drinking and smoking. Reducing the prevalence of drinking and smoking combined with dual addiction in the community requires identifying significant model parameters that have an influence on the dual addiction model. We must do a sensitivity analysis of the model parameters using the values listed in Table 4 below, together with the dual addiction effective reproduction number, represented by $\mathcal{R}_{0AS}$, using the following widely recognized approach.

**Definition:** Let $a$ be an arbitrary model parameter incorporated in the dual addiction model effective reproduction number $\mathcal{R}_{0AS}$, and then the forward sensitivity index formula is defined by $SI_a^{\mathcal{R}_{0AS}} = \frac{\partial \mathcal{R}_{0AS}}{\partial a} \times \frac{a}{\mathcal{R}_{0AS}}$ [23, 33, 45].

Applying the dual addiction dissemination model parameters stated in Table 4, we have demined that $\mathcal{R}_{0A} = 1.74 > 1$ and $\mathcal{R}_{0S} = 1.53 > 1$ and also we have computed the values $\mathcal{R}_{0AS} = \max\{\mathcal{R}_{0A}, \mathcal{R}_{0S}\} = \mathcal{R}_{0A} = 1.74 > 1$. And also we computed the sensitivity indices as:

1. $SI_{\beta_1}^{\mathcal{R}_{0A}} = \frac{\partial \mathcal{R}_{0A}}{\partial \beta_1} \times \frac{\beta_1}{\mathcal{R}_{0A}} = 1 > 0.$

2. $SI_{\alpha_1}^{\mathcal{R}_{0A}} = \frac{\mu}{(\mu + \alpha_1)} > 0.$

3. $SI_{\alpha_2}^{\mathcal{R}_{0A}} = \frac{\alpha_2 [1 - (\mu + \alpha_2)]}{(\alpha_2 + \varepsilon_1 + \mu + d_A)(\mu + \alpha_2)}.$

4. $SI_\pi^{\mathcal{R}_{0A}} = \frac{[-\mu + \rho_1 \mu]\pi}{(\mu + \alpha_1)(\alpha_2 + \varepsilon_1 + \mu + d_A)(\mu + d_{PA})(\delta + \mu)[\mu(1 - \pi) + \rho_1(\delta + \pi\mu)]}.$

5. $SI_{\rho_1}^{\mathcal{R}_{0A}} = \frac{\rho_1(\delta + \pi\mu)}{(\mu + \alpha_1)(\alpha_2 + \varepsilon_1 + \mu + d_A)(\mu + d_{PA})(\delta + \mu)[\mu(1 - \pi) + \rho_1(\delta + \pi\mu)]}.$

6. $SI_\delta^{\mathcal{R}_{0A}} = \frac{[\mu\rho_1(1 - \pi) - \mu(1 - \pi)]\delta}{(\delta + \mu)^2 [\mu(1 - \pi) + \rho_1(\delta + \pi\mu)]}.$

7. $SI_{\varepsilon_1}^{\mathcal{R}_{0A}} = \frac{-(\mu + \alpha_1)\varepsilon_1}{(\mu + \alpha_1)(\alpha_2 + \varepsilon_1 + \mu + d_A)}.$

8. $SI_{d_A}^{\mathcal{R}_{0A}} = \frac{-d_A}{(\alpha_2 + \varepsilon_1 + \mu + d_A)}.$

**Table 4. The dual addiction model parameters values used for numerical simulation.**

| Parameters | Values ($year^{-1}$) | Sources |
|---|---|---|
| $\mu$ | 0.0135 | [4] |
| K | 20.00 | [2] |
| $\pi$ | 0.5000 | Assumed |
| $\delta_1,\delta_2,\delta_3,\rho_1,\rho_2,\rho_3,$ | 1.1, 1.1,1.1,1.2,1.2,1.2 | Assumed |
| $\varepsilon_1$ | 0.223 | [29] |
| $\varepsilon_2$ | 0.0100 | [2] |
| $\varepsilon_3$ | 0.3130 | Assumed |
| $\beta_1$ | 0.7500 | [10] |
| $\beta_2$ | 0.3800 | [3] |
| $\alpha_1$ | 0.0002, | [28] |
| $\gamma_1$ | 0.0100 | [6] |
| $\alpha_1$ | 0.0003 | Assumed |
| $\alpha_2$ | 0.5600 | [29] |
| $\delta$ | 0.2100 | Assumed |
| $\theta_1$ | 1.05 | [37] |
| $\theta_2$ | 1.05 | [37] |
| $\alpha$ | 0.5400 | Assumed |
| $\gamma_2$ | 0.3210 | Assumed |
| $d_A$ | 0.075 | [37] |
| $d_S$ | 0.035 | [37] |
| $d_{PA}$ | 0.091 | Assumed |
| $d_{PS}$ | 0.87 | Assumed |
| $d_C$ | $d_A+d_S = 0.11$ | [37] |
| $d_{SA}$ | 0.23 | Assumed |

9. $SI^{\mathcal{R}_{0A}}_{d_{PA}} = \frac{-d_{PA}}{\mu+d_{PA}}$.

Based on the model parameter values described in Table 4 below we have determined the sensitivty index values represented by Table 3 below.

Based on the sensitivity indices described in Table 3 aboe we have the following diagrams that show the graphical representations of the values represented in Table 3 above.

## 4. The optimal control problem and its analyses

In this section, based on the smoking and alcohol dual addiction dissemination model (4) parameters sensitivity indices described in Table 3 above and the diagram illustrated by Fig 2 above we re-constructed the optimal control problem (15) by considering the bounded, Lebesgue integrable control functions, denoted by w = ($w_1,w_2,w_3,w_4,w_5,w_6$) such that

- The control $w_1(t)$ represent efforts to protect individuals against smoking addiction by educating them about the impact of smoking dissemination in the community.

- The control function $w_2(t)$ represent efforts to protect individuals against alcohol addiction by educating them about the impact of alcohol dissemination in the community.

- The control function $w_3(t)$ represent efforts to protect individuals against dual addiction by educating them about the impact of dual addiction dissemination in the community.

- The control function $w_4(t)$ represent the treatment strategies (punishment and psychological treatment) given for individuals who are smoking addicted but not for permanently addicted individuals.

- The control function $w_5(t)$ represent the treatment strategies (punishment and psychological treatment) given for individuals who are alcohol addicted but not for permanently addicted individuals.

- The control function $w_6(t)$ represent the treatment strategies (punishment and psychological treatment) given for individuals who are dually addicted but not for permanently addicted individuals.

The main objective is to find the optimal control values $w^* = (w_1^*, w_2^*, w_3^*, w_4^*, w_5^*, w_6^*)$ of the proposed control measures $w = (w_1, w_2, w_3, w_4, w_5, w_6)$ such that the associated state trajectories $S^*, P^*, E_S^*, I_S^*, P_S^*, E_A^*, I_A^*, P_A^*, C_{SA}^*, P_{SA}^*, T^*$ are solutions of the system (4) in the intervention time interval $[0, T_F]$ with initial conditions given in (5) and also to minimize the objective functional.

Thus, implementation of the right rehabilitation and protection policies for smoking and alcohol dual addiction, smoking or alcohol single addiction in a community is fundamental to improve the rehabilitation and treatment period such that $0 \leq w_1, w_2, w_3, w_4, w_5, w_6 \leq 1$.

The optimal control problem system for the smoking and alcohol dual addiction dissemination model (4) can be re-formulated as:

$$\frac{dS}{dt} = (1 - \pi)\mathrm{K} - (1 - w_1)\lambda_S S - (1 - w_2)\lambda_A S - w_3 \delta S - \mu S,$$

$$\frac{dP}{dt} = \pi \mathrm{K} + w_3 \delta S - (1 - w_2)\rho_1 \lambda_A P - (1 - w_1)\delta_1 \lambda_S P - \mu P,$$

$$\frac{dE_S}{dt} = (1 - w_1)\lambda_S S + (1 - w_1)\delta_1 \lambda_S P - (\mu + \gamma_1)E_S,$$

$$\frac{dI_S}{dt} = \gamma_1 E_S - w_4 \varepsilon_2 I_S - \gamma_2 I_S - (1 - w_4)\rho_3 \lambda_A I_S - (\mu + d_S)I_S,$$

$$\frac{dP_S}{dt} = \gamma_2 I_S - (1 - w_4)\rho_2 \lambda_A P_S - (\mu + d_{PS})P_S, \tag{15}$$

$$\frac{dE_A}{dt} = (1 - w_2)\lambda_A S + (1 - w_2)\rho_1 \lambda_A P - (\mu + \alpha_1)E_A,$$

$$\frac{dI_A}{dt} = \alpha_1 E_A - \alpha_2 I_A - w_5 \varepsilon_1 I_A - (1 - w_1)\delta_2 \lambda_S I_A - (\mu + d_A)I_A,$$

$$\frac{dP_A}{dt} = \alpha_2 I_A - (1 - w_1)\delta_3 \lambda_S P_A - (\mu + d_{PA})P_A,$$

$$\frac{dC_{SA}}{dt} = (1 - w_1)\delta_2 \lambda_S I_A + (1 - w_1)\delta_3 \lambda_S P_A + (1 - w_2)\rho_3 \lambda_A I_S + (1 - w_2)\rho_2 \lambda_A P_S - (\alpha + w_6 \varepsilon_3 + \mu + d_C)C_{SA},$$

$$\frac{dP_{SA}}{dt} = \alpha C_{SA} - (\mu + d_{SA})P_{SA},$$

$$\frac{dT}{dt} = w_6 \varepsilon_3 C_{SA} + w_5 \varepsilon_1 I_A + w_4 \varepsilon_2 I_S - \mu T,$$

with initial population given by

$$S(0) > 0, P(0) \geq 0, E_S(0) \geq 0, I_S(0) \geq 0, P_S(0) \geq 0, E_A(0) \geq 0, I_A(0) \geq 0, P_A(0) \geq 0, C_{SA}(0)$$
$$\geq 0, P_{SA}(0) \geq 0, \text{ and } T(0) \geq 0. \tag{16}$$

The corresponding optimal control problem objective functional is represented by

$$J(w_1, w_2, w_3, w_4, w_5, w_6) = \int_0^{T_F} \left( D_1 I_S + D_2 P_S + D_3 I_A + D_4 P_A + D_5 C_{SA} + D_6 P_{SA} + \frac{1}{2} \sum_{i=1}^{6} \psi_i w_i^2 \right) dt, \tag{17}$$

where $I(S, P, E_S, I_S, P_S, E_A, I_A, P_A, C_{SA}, P_{SA}, T, w_1, w_2, w_3, w_4, w_5, w_6) = D_1 I_S + D_2 P_S + D_3 I_A + d_4 P_A + D_5 C_{SA} + D_6 P_{SA} + \frac{\psi_1}{2} w_1^2 + \frac{\psi}{2} w_2^2 + \frac{\psi_3}{2} w_3^2 + \frac{\psi_4}{2} w_4^2 + \frac{\psi_5}{2} w_5^2 + \frac{\psi_6}{2} w_6^2$, measures the current cost at time $t$, $T_F$ is the final time, the coefficients $D_1, D_2, D_3, D_4, D_5$ and $D_6$ are positive weight constants and $\frac{\psi_1}{2}, \frac{\psi_2}{2}, \frac{\psi_3}{2}, \frac{\psi_4}{2}, \frac{\psi_5}{2}$ and $\frac{\psi_6}{2}$ are the measure of relative costs of interventions associated with the control function $w_1, w_2, w_3, w_4, w_5$ and $w_6$, respectively and also balances the units of integrand. In the cost functional, the term $D_1 I_S$ refer to the cost related to smoking addiction towards a community, the term $D_2 P_S$ refer to the cost related to permanent smoking addicted group, the term $D_3 I_A$ refer to the cost related to alcohol addiction towards a community, the term $D_4 P_A$ describe the cost related to permanent alcohol addicted group, the term $D_5 C_{SA}$ describe the cost related to smoking and alcohol dual addicted groups towards a community, and the term $D_6 P_{SA}$ describe the cost related to permanent smoking and alcohol dual addicted group towards a community. And the cost functional $J$ corresponds to the total cost due to smoking and alcohol dual addiction and its control strategies.

The set of admissible control functions are defined by

$$\Omega_w = \{w_1(t), w_2(t), w_3(t), w_4(t), w_5(t), w_6(t) \in L^6 : 0 \leq w_1, w_2, w_3, w_4, w_5, w_6 \leq 1, t \in [0, T_F]\}. \tag{18}$$

More precisely, we need an optimal control pair

$$J(w_1^*, w_2^*, w_3^*, w_4^*, w_5^*, w_6^*) = \min_{\Omega_w} J(w_1, w_2, w_3, w_4, w_5, w_6). \tag{19}$$

## 4.1 Characterization of optimal control

Based on the fundamental criteria called the Pontryagin's Maximum principle used in [39, 40, 46], if $w^*(t) \in \Omega_w$ is optimal control for dynamical system (15) with initial population stated in Eq (16) and an optimal control pair stated in Eq (19) with fixed final time $T_F$, then there exists an absolutely continuous mapping

$$f : [0, T_F] \longrightarrow \mathbb{R}^{11}, f = (f_1, f_2, f_3, f_4, f_5, f_6, f_7, f_8, f_9, f_{10}, f_{11}),$$ is called the adjoint vector, such that

1. The Hamiltonian function is defined as

$$H = D_1 I_S + D_2 P_S + D_3 I_A + D_4 P_A + D_5 C_{SA} + D_6 P_{SA} + \frac{1}{2} \sum_{i=1}^{6} \psi_i w_i^2 + \sum_{i=1}^{11} f_i \chi_i, \tag{20}$$

where $\chi_i$ stands for the right hands side of the optimal control problem model (15) which is the $i^{th}$ state variable equation.

2. The control system

$$\frac{dS}{dt} = \frac{\partial H}{\partial f_1}, \frac{dP}{dt} = \frac{\partial H}{\partial f_2}, \frac{dE_S}{dt} = \frac{\partial H}{\partial f_3}, \frac{dI_S}{dt} = \frac{\partial H}{\partial f_4}, \frac{dP_S}{dt} = \frac{\partial H}{\partial f_5}, \frac{dE_A}{dt} = \frac{\partial H}{\partial f_6}, \frac{dI_A}{dt} = \frac{\partial H}{\partial f_7}, \frac{dP_A}{dt}$$

$$= \frac{\partial H}{\partial f_8}, \frac{dC_{SA}}{dt} = \frac{\partial H}{\partial f_9}, \frac{dP_{SA}}{dt} = \frac{\partial H}{\partial f_{10}}, \frac{dT}{dt} = \frac{\partial H}{\partial f_{11}}. \tag{21}$$

3. The adjoint system

$$\frac{df_1}{dt} = -\frac{\partial H}{\partial S}, \frac{df_2}{dt} = -\frac{\partial H}{\partial P}, \frac{df_3}{dt} = -\frac{\partial H}{\partial E_S}, \frac{df_4}{dt} = -\frac{\partial H}{\partial I_S}, \frac{df_5}{dt} = -\frac{\partial h}{\partial P_S}, \frac{df_6}{dt} = -\frac{\partial H}{\partial E_A}, \frac{df_7}{dt}$$

$$= -\frac{\partial H}{\partial I_A}, \frac{df_8}{dt} = -\frac{\partial H}{\partial P_A}, \frac{df_9}{dt} = -\frac{\partial H}{\partial C_{SA}}, \frac{df_{10}}{dt} = -\frac{\partial H}{\partial P_{SA}}, \frac{df_{11}}{dt} = -\frac{\partial H}{\partial T}. \tag{22}$$

And the optimality condition

$$H(\mathbb{E}^*, w, f^*) = \min_{w \in \Omega_w}(\mathbb{E}^*, w^*, f^*), \tag{23}$$

holds for almost all $t \in [0, T_F]$.

4. Moreover, the transversality condition

$$f_i(T_F) = 0, i = 1, 2, 3, \ldots, 11 \tag{24}$$

also holds true. In the next result, we discuss characterization of optimal controls and adjoint variables.

**Theorem 9**: Let $w^* = (w_1^*, w_2^*, w_3^*, w_4^*, w_5^*, w_6^*)$ be the optimal control strategies and $(S^*(t), P^*(t), E_S^*(t), I_S^*(t), P_S^*(t), E_A^*(t), I_A^*(t), P_A^*(t), C_{SA}^*(t), P_{SA}^*(t), T^*(t))$ be the associated unique optimal solutions of the optimal control problem (15) with initial population given by (16) and objective functional (17) with fixed final time $T_F$ (18). Then there exists adjoint function $f_i^*(t), i = 1, \ldots, 11$ satisfying the following canonical equations:

$$\frac{df_1}{dt} = (1 - w_1)\lambda_S^*(f_1 - f_3) + (1 - w_2)\lambda_A^*(f_1 - f_6) + w_3\delta(f_1 - f_2) + \mu f_1,$$

$$\frac{df_2}{dt} = (1 - w_2)\rho_1\lambda_A^*(f_2 - f_6) + (1 - w_1)\delta_1\lambda_S^*(f_2 - f_3) + \mu f_2,$$

$$\frac{df_3}{dt} = \mu f_3 + \gamma_1(f_3 - f_4),$$

$$\frac{df_4}{dt} = -D_1 + (1 - w_1)\frac{\beta_2}{N^*}\left[S^*(f_1 - f_3) + \delta_1 P^*(f_2 - f_3) + I_A^*\delta_2(f_7 - f_9) + P_A^*\delta_3(f_8 - f_9)\right]$$
$$+ \gamma_2(f_4 - f_5) + (1 - w_2)(f_4 - f_9)\rho_3\lambda_A^* + (w_5\varepsilon_2 + \mu)f_4,$$

$$\frac{df_5}{dt} = -D_2 + (1 - w_1)\frac{\beta_2}{N^*}\left[N^*(f_1 - f_3) + \delta_1 P^*(f_2 - f_3) + \delta_2 I_A^*(f_7 - f_9) + \rho_2 P_A^*(f_8 - f_9)\right]$$
$$+ (1 - w_2)\rho_2\lambda_A^*(f_5 - f_9),$$

$$\frac{df_6}{dt} = \mu f_6 + \alpha_1 (f_6 - f_7),$$

$$\frac{df_7}{dt} = -D_3 + (1 - w_2)\frac{\beta_1}{N^*}\left[S^*(f_1 - f_6) + \rho_1 P^*(f_2 - f_6) + \rho_3 I_S^*(f_4 - f_9) + \rho_2 P^*(f_5 - f_9)\right]$$
$$+ (1 - w_1)\delta_2 \lambda_S^*(f_7 - f_9) + \alpha_2(f_7 - f_8) + w_4 \varepsilon_1(f_7 - f_{11}) + \mu f_7,$$

$$\frac{df_8}{dt} = -D_4 + (1 - w_2)\frac{\beta_1}{N^*}\left[S^*(f_1 - f_6) + \rho_1 P^*(f_2 - f_6) + \rho_3 I_S^*(f_4 - f_9) + \rho_2 P_A^*(f_5 - f_5)\right]$$
$$+ (1 - w_1)\delta_3 \lambda_S^*(f_8 - f_9) + \mu f_8,$$

$$\frac{df_9}{dt} = -D_5 + (1 - w_1)\frac{\beta_2 \theta_1}{N^*}\left[S^*(f_1 - f_3) + \delta_1 P^*(f_2 - f_3) + \delta_2 I_A^*(f_7 - f_9) + \rho_3 P_A^*(f_8 - f_9)\right]$$
$$+ (1 - w_2)\frac{\beta_1 \alpha_1}{N^*}\left[S^*(f_1 - f_6) + \rho_3 I_S^*(f_4 - f_9) + \rho_1 P^*(f_2 - f_6) + \rho_2 P_A^*(f_5 - f_9)\right] + \mu f_9$$
$$+ \alpha(f_9 - f_{10}) + w_6 \varepsilon_3(f_9 - f_{11}),$$

$$\frac{df_{10}}{dt} = -D_6 + (1 - w_1)\frac{\beta_2 \theta_1}{S^*}\left[S^*(f_1 - f_3) + \delta_1 P^*(f_2 - f_3) + \delta_2 I_A^*(f_7 - f_9) + \delta_3 P_A^*(f_8 - f_9)\right]$$
$$+ (1 - w_2)\frac{\beta_1 \alpha_1}{N^*}\left[S^*(f_1 - f_6) + \rho_3 I_S^*(f_4 - f_9) + \rho_1 P^*(f_2 - f_6) + \rho_2 P_A^*(f_5 - f_9)\right] + \mu f_{10},$$

$$\frac{df_{11}}{dt} = \mu f_{11},$$

with transiversality conditions

$$f_i^*(T_F) = 0, i = 1, 2, \ldots, 11. \tag{25}$$

Moreover, the corresponding optimal control strategies $w_1^*(t), w_2^*(t), w_3^*(t), w_4^*(t), w_5^*(t)$ and $w_6^*(t)$ are determined as

$$w_1^*(t) = \max\left\{0, min\left\{\frac{\lambda_S^*(S^*(f_3 - f_1) + \delta_1 P^*(f_3 - f_2) + \delta_2 I_A^*(f_9 - f_7) + \delta_3 P_A^*(f_9 - f_8))}{\psi_1}, 1\right\}\right\},$$

$$w_2^*(t) = \max\left\{0, min\left\{\frac{\lambda_A^*[S^*(f_6 - f_1) + \rho_1 P^*(f_6 - f_2) + \rho_3 I_S^*(f_9 - f_4) + \rho_2 P_S^*(f_9 - f_5))}{\psi_2}, 1\right\}\right\},$$

$$w_3^*(t) = \max\left\{0, min\left\{\frac{\delta S^*(f_1 - f_2)}{\psi_3}, 1\right\}\right\}, \tag{26}$$

$$w_4^*(t) = \max\left\{0, min\left\{\frac{\varepsilon_1 I_A^*(f_7 - f_{11})}{\psi_4}, 1\right\}\right\},$$

$$w_5^*(t) = \max\left\{0, min\left\{\frac{\varepsilon_2 I_S^*(f_4 - f_{11})}{\psi_5}, 1\right\}\right\},$$

$$w_6^*(t) = \max\left\{0, min\left\{\frac{\varepsilon_3 C_{SA}^*(f_9 - f_{11})}{\psi_6}, 1\right\}\right\}.$$

## 5. Numerical simulations

Various differential equations are challenging to solve analytically. In this scenario, a numerical simulation of the dynamical system is required, hence using MATLAB ode45 solver with fourth order Runge-Kutta numerical methods like used in [47], we employed numerical simulations to verify the backward bifurcation phenomenon for the dual addiction dynamics, to investigate the smoking and alcoholism dual addiction model (4) solutions behavior whenever its effective reproduction number is greater than unity($\max\{\mathcal{R}_{0A}, \mathcal{R}_{0S}\} = \mathcal{R}_{0A} = 1.74{>}1$). Also to examine the impacts of the proposed time dependent optimal control strategies on the dual addiction dissemination dynamics in the community using parameter values described in Table 4 below.

### 5.1 Simulation of the backward bifurcation

The alcohol addiction sub-model and the smoking addiction sub-model backward bifurcations phenomena are revealed by numerical simulations, as seen in Figs 3 and 4, respectively. The outcomes of the simulations confirm that, in situations where the corresponding effective reproduction numbers ($\mathcal{R}_{0A}{<}1$, and $\mathcal{R}_{0S}{<}1$) is smaller than unity, alcohol dominance and alcohol-free equilibrium points as well as smoking dominance and smoking-free equilibrium points coexist respectively. The smoking and drinking dual addiction model also demonstrates the phenomena of backward bifurcation if its effective reproduction number is less than unity since the dual addiction effective reproduction number is the higher of the two single addiction sub-models. Thus, even if the dual addiction dissemination model effective reproduction

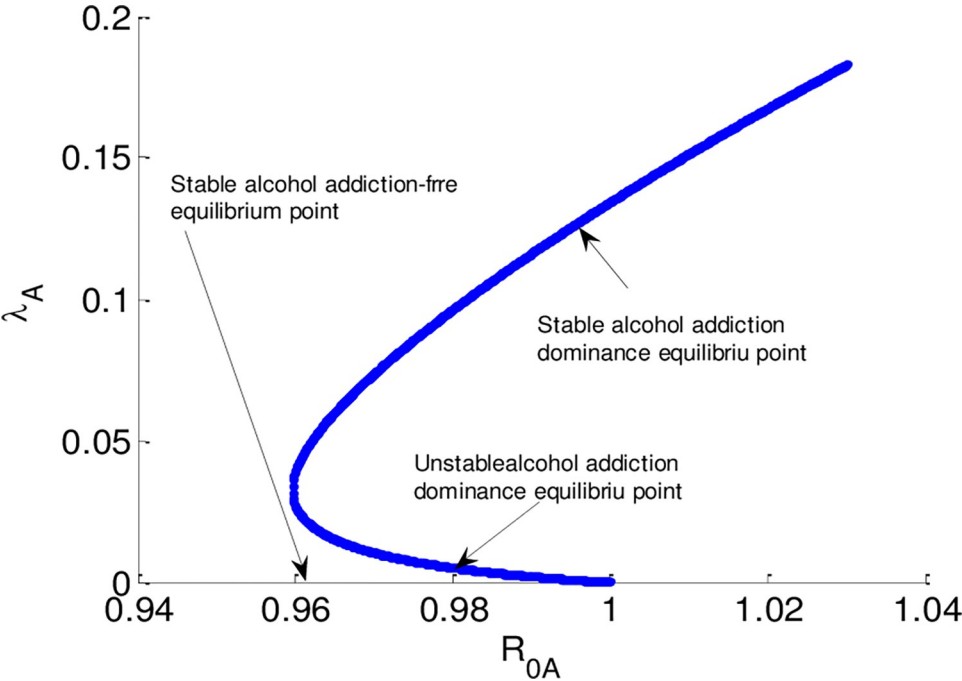

**Fig 3. Bifurcation curve of alcohol addiction sub-model.**

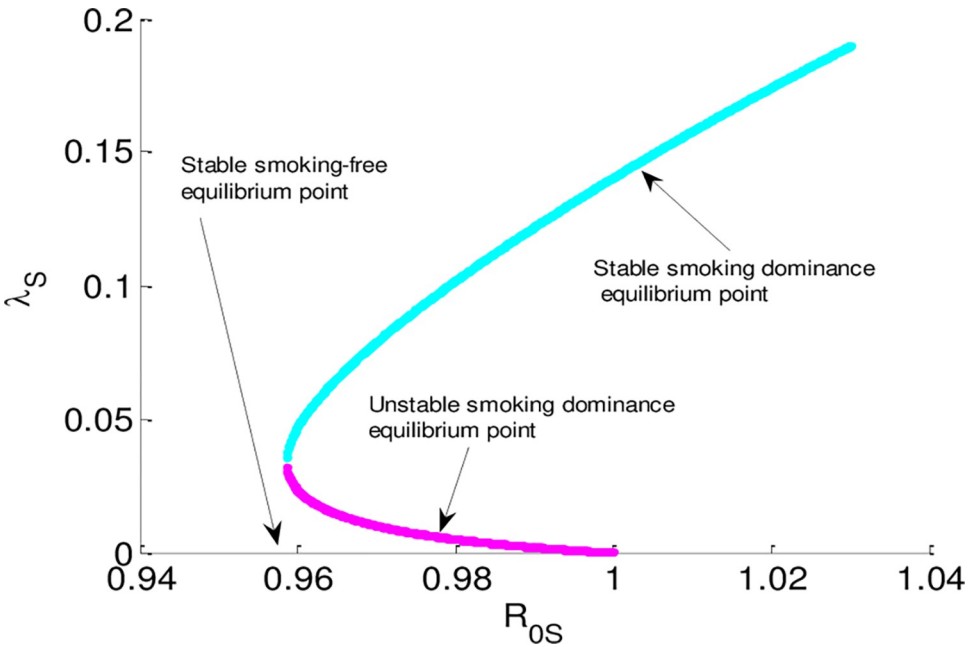

**Fig 4. Bifurcation curve of smoking addiction sub-model.**

number is less than unity the smoking and alcoholism dual addiction disseminates in the community.

## 5.2 Numerical simulation of the dual addiction model (4)

In this sub-section of the study, we applied the dual addiction model parameter values described in Table 4 above we carried out the numerical simulation for the smoking and alcoholism dual addiction dissemination model (4) and we found the simulation curves of the dynamical system (4) solutions given by Fig 5 whenever $\max\{\mathcal{R}_{0A}, \mathcal{R}_{0S}\} = \mathcal{R}_{0A} = 1.74 > 1$. According to the result illustrated by Fig 5 we observed that the model (4) solution curves attracted towards the dual addiction model persistence equilibrium point if $\max\{\mathcal{R}_{0A}, \mathcal{R}_{0S}\} = \mathcal{R}_{0A} = 1.74 > 1$, and thus, from the trajectories illustrated by Fig 5 above one can justify that the smoking and alcoholism dual addiction disseminate throughout the community uniformly.

## 5.3 Simulation of the optimal control problem model (15)

Since numerical simulation provides the qualitative analysis with an illustrated perspective of the mathematical model, it is a crucial tool in the mathematical modeling of dynamics in real-world issues. Therefore, by simulating nine alternative combinations of control techniques, we execute the numerical simulation outcomes of the investigation in this study. With the assumptions $\psi_1 = \psi_2 = \psi_3 = \psi_4 = \psi_5 = \psi_6 = 10$, and $D_1 = D_2 = D_3 = D_4 = D_5 = D_6 = 7$ and different initial conditions, the numerical solutions are illustrated using MATLAB ode45 solver with fourth order Runge-Kutta forward–backward sweep numerical method (due to its convergence and stability) verified by Lenhart and Worksman criteria stated in [45] and by applying data stated in Table 4. Numerical simulations are conducted on the optimum control issue (15) in order to determine the best ways to lessen the prevalence of alcohol and tobacco addiction in the society. Case 1: The simultaneous use of three preventative measures (education) against alcohol, smoking, and dual addictions (Strategy A), as well as three remedial measures

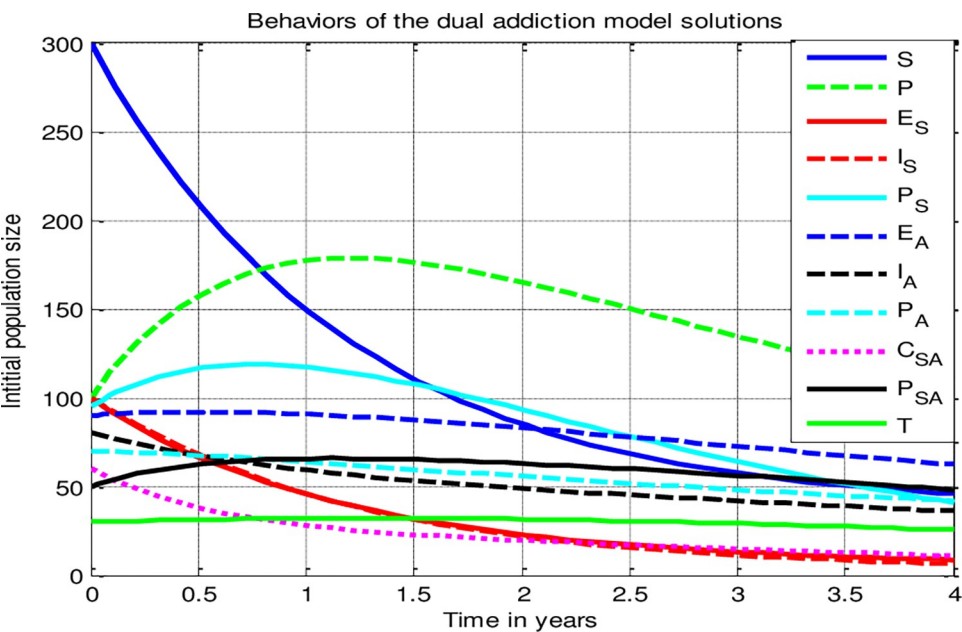

**Fig 5. Trajectories of the dual addiction model (4) solutions whenever $\mathcal{R}_{0AS}>1$.**

(rehabilitation) for these addictions (Strategy B). Case 2: Implementing education against alcohol, smoking, and dual addictions along with a smoking treatment measure concurrently (Strategy C); implementing education against alcohol, smoking, and dual addictions along with an alcoholism treatment measure concurrently (Strategy D); and implementing education against alcohol, smoking, and dual addictions along with a dual addiction treatment measure concurrently (Strategy E). Case 3: Using education as a safeguard against alcohol, smoking, and dual addictions and implementing alcohol and smoking rehabilitation measures

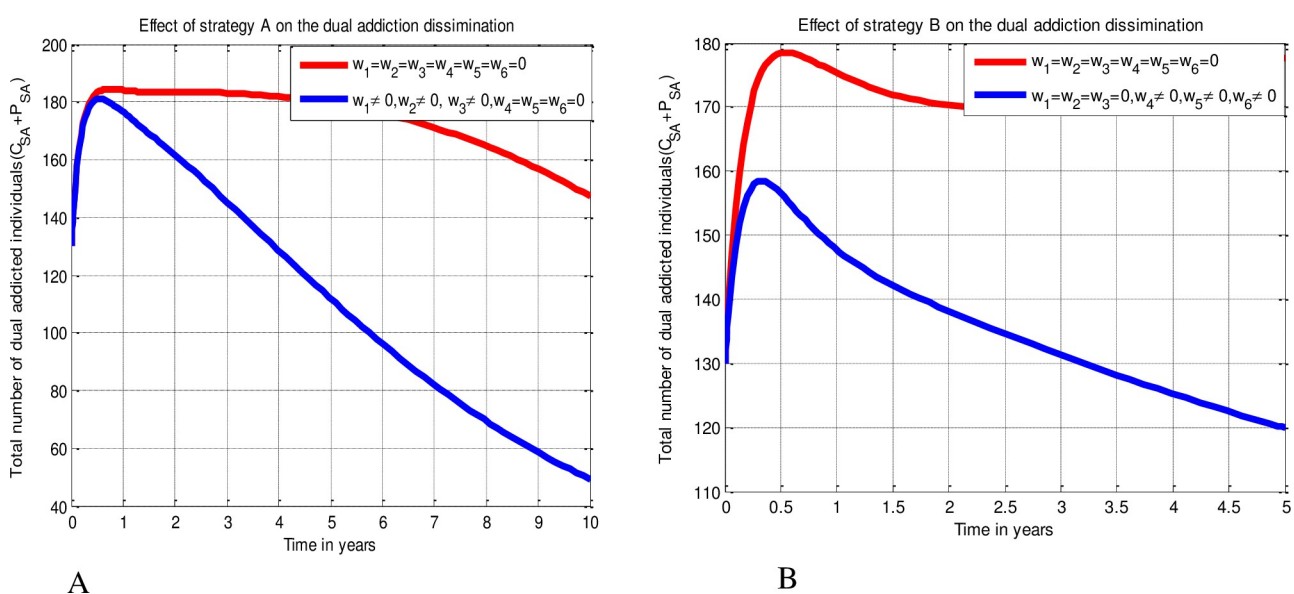

**Fig 6. Impacts of Strategy A and Strategy B on the dual addiction dissemination respectively.**

concurrently (Strategy F); using education as a safeguard against alcohol, smoking, and dual addictions and implementing smoking and dual addictions rehabilitation measures concurrently (Strategy G); and using education as a safeguard against alcohol, smoking, and dual addictions and alcoholism and dual addiction treatment measures concurrently (Strategy H). Case 4: Applying all safeguards and rehabilitative control procedures at the same time (Strategy I). Iterated below as

**Case 1 (Implementation of three control measures).** Strategy A: Implementing protection strategies ($w_1 \neq 0$, $w_2 \neq 0$, $w_3 \neq 0$) simultaneously.

Strategy B: Implementing treatment strategies ($w_4 \neq 0$, $w_5 \neq 0$, $w_6 \neq 0$) simultaneously.

**Case 2 (Implementation of four control measures).** Strategy C: Implementing protection strategies ($w_1 \neq 0$, $w_2 \neq 0$, $w_3 \neq 0$) and treatment strategy ($w_4 \neq 0$).

Strategy D: Implementing protection strategies ($w_1 \neq 0$, $w_2 \neq 0$, $w_3 \neq 0$) and treatment strategy ($w_5 \neq 0$).

Strategy E: Implementing protection strategies ($w_1 \neq 0$, $w_2 \neq 0$, $w_3 \neq 0$) and treatment strategy ($w_6 \neq 0$).

**Case 3 (Implementation of five control measures).** Strategy F: Implementing protection strategies ($w_1 \neq 0$, $w_2 \neq 0$, $w_3 \neq 0$) and treatment strategies ($w_4 \neq 0$, $w_5 \neq 0$).

Strategy G: Implementing protection strategies ($w_1 \neq 0$, $w_2 \neq 0$, $w_3 \neq 0$) and treatment strategies ($w_4 \neq 0$, $w_6 \neq 0$).

Strategy H: Implementing protection strategies ($w_1 \neq 0$, $w_2 \neq 0$, $w_3 \neq 0$) and treatment strategies ($w_5 \neq 0$, $w_6 \neq 0$).

**Case 4 (Implementation of all control measures).** Strategy I: Implementing all proposed strategies ($w_1 \neq 0$, $w_2 \neq 0$, $w_3 \neq 0$, $w_4 \neq 0$, $w_5 \neq 0$, $w_6 \neq 0$) simultaneously.

**5.3.1 Effect of the triple control measures on the dual addiction.** The numerical simulation results of the optimal control system (15) when the strategies A and B are presented in Fig 6 above respectively. It can be seen from Fig 6(A) that implementing Strategy A tends to minimize the number of smoking and alcoholism dual addiction greatly. However, the number of smoking and alcoholism dual addicted individuals is high whenever no intervention strategies put in place against the dual addiction dissemination. The result shown us the impact of implementing the smoking, alcoholism, and smoking and alcoholism dual addiction educations protection measures simultaneously (implementing $w_1 \neq 0$, $w_2 \neq 0$, $w_3 \neq 0$, $w_4 = w_5 = w_6 = 0$ simultaneously) on the dual addiction dissemination dynamics in the community. Thus, implementing the education protection measures simultaneously (i.e., implementing Strategy A) has a significance role to reduce the dual addiction dissemination in the community as compared with the simulation result without implementing any proposed control strategies. Similarly, the simulation curve given by Fig 6(B) reveals the possible effects (impacts) of smoking addiction, alcoholism addiction, and smoking and alcoholism dual addiction improvement (rehabilitation) measures simultaneously, i.e., $w_1 = w_2 = w_3 = 0$, $w_4 \neq 0$, $w_5 \neq 0$, $w_6 \neq 0$ denoted by Strategy B on the dual addiction dissemination dynamics in the community. The result of the simulation given by Fig 6(B) shows that implementing the improvement (rehabilitation) control measures simultaneously (i.e., implementing Strategy B) has a significance role to reduce the dual addiction dissemination dynamics in the community as compared with the simulation result without implementing any proposed control strategies. From the simulation results explained implementing the Strategy A (i.e., implementing protection measures simultaneously is more effective to reduce the dual addiction dissemination than implementing Strategy B (i.e., implementing treatment measures simultaneously).

**5.3.2 Effect of four control measures on the dual addiction.** The potential repercussions of applying all four control measures at once are shown by the simulation curves illustrated in Fig 7. Fig 7(C) demonstrated the effects of concurrently implementing the smoking

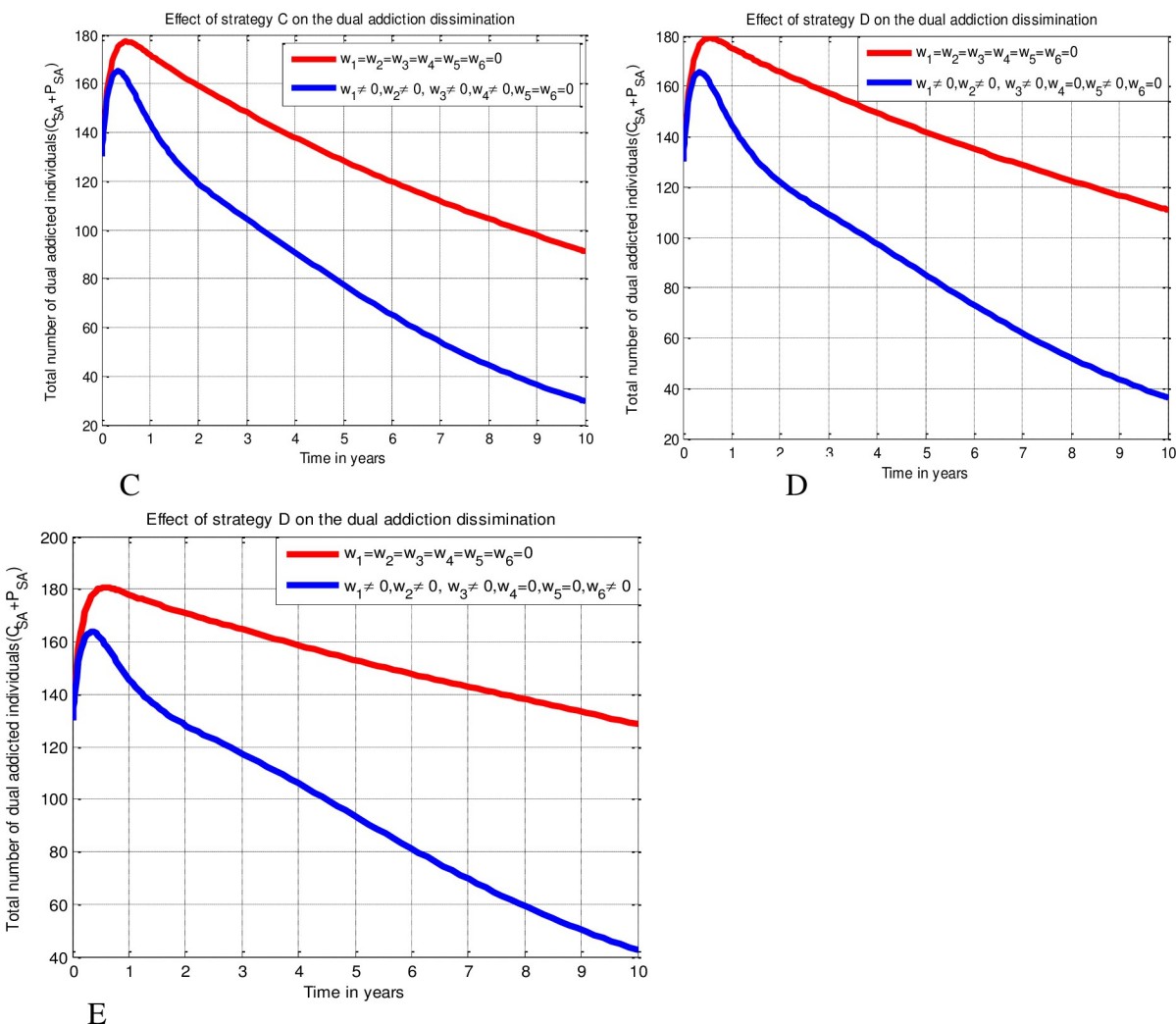

**Fig 7.** Impacts of Strategy C, Strategy D, and Strategy E on the dual addiction dissemination respectively.

improvement (rehabilitation) control measure and the alcoholism addiction and smoking addiction protection measures, as described by Strategy C, on the dual addiction dissemination dynamics in the community. As compared to the simulation cure without implementing any suggested strategies, the simulation result shown in Fig 7(C) indicates that implementing protection control measures and smoking improvement (rehabilitation) control measures simultaneously ($w_1 \neq 0$, $w_2 \neq 0$, $w_3 \neq 0$, $w_4 \neq 0$ or Strategy C) has a significant role to reduce the dual addiction dissemination dynamics in the community. Additionally, as shown by Fig 7 (D) in the simulation curve, the potential impacts (effects) of alcoholism addiction, smoking addiction, and protection measures against both alcoholism and smoking addiction as well as alcohol addiction improvement (rehabilitation) control measures simultaneously ($w_1 \neq 0$, $w_2 \neq 0$, $w_3 \neq 0$, $w_5 \neq 0$ or Strategy D) on the dual addiction dissemination problem in the community are revealed. As compared to the simulation cure without implementing any suggested strategies, the simulation result shown in Fig 7(D) indicates that implementing protection and alcoholism addiction improvement (rehabilitation) control measures simultaneously (Strategy D) has a significant role to reduce the dual addiction dissemination in the community. In a

similar vein, the simulation curve depicted in Fig 7(E) shows the potential impacts (effects) of alcoholism addiction, smoking addiction, and simultaneous implementation of protection and improvement (rehabilitation) control measures for both alcoholism and smoking addiction (i.e., implementing Strategy E) on the dual addiction dissemination problem in the community. The simulation result shown in Fig 7(E) demonstrates that, in comparison to the simulation result without implementing any suggested control strategies, implementing all protection measures and dual addiction improvement (rehabilitation) control measures simultaneously (i.e., implementing Strategy E) has a significant role to reduce the dual addiction dissemination in the community. Ultimately, if we do not take into account the cost-effectiveness analysis of each of these control strategies, we compare the results of Fig 7C–7E and find that the most effective strategy to reduce the prevalence of alcoholism and smoking dual addiction in the community is to implement Strategy E, which entails implementing all of the protection measures and dual addiction rehabilitation measures simultaneously ($w_1 \neq 0$, $w_2 \neq 0$, $w_3 \neq 0$, $w_6 \neq 0$).

**5.3.3 Effect of five control measures on the dual addiction.** The potential impacts of implementing protection measures and smoking and alcoholism single addictions improvement (rehabilitation) control measures simultaneously ($w_1 \neq 0$, $w_2 \neq 0$, $w_3 \neq 0$, $w_4 \neq 0$, $w_5 \neq 0$ described by strategy F) on the dual addiction dissemination problem in the community are shown by the simulation curves depicted in Fig 8. The simulation result shown in Fig 8(F) demonstrates that, when compared to the simulation cure without implementing any suggested control measures, implementing protection measures and smoking and alcoholism

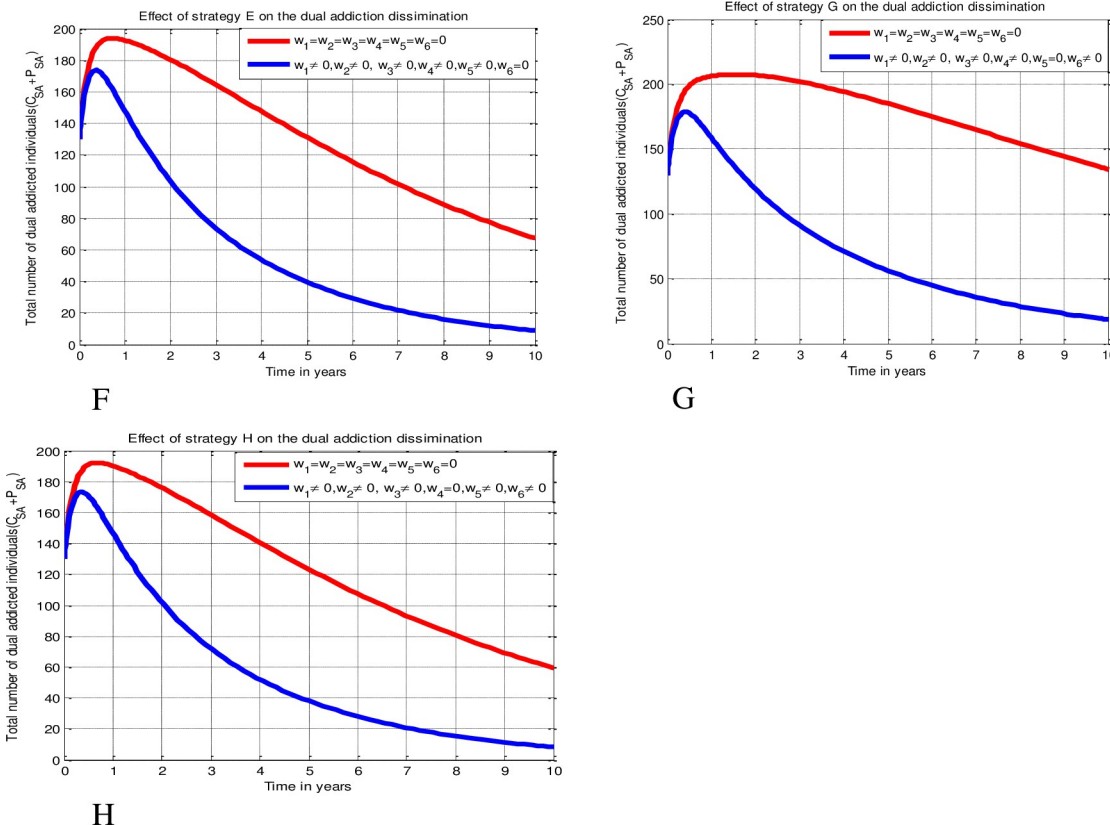

**Fig 8.** Impacts of Strategy F, Strategy G, and Strategy H on the dual addiction dissemination respectively.

single addictions improvement (rehabilitation) control measures simultaneously (i.e., implementing Strategy F) has a significant role to reduce the dual addiction dissemination in the community. Additionally, the numerical simulation curve depicted in Fig 8(G) shows the potential outcomes (impacts) of putting protective measures in place as well as the improvement of multiple addictions and smoking-only addiction (rehabilitation). The simulation result shown in Fig 8(G) demonstrates that, in comparison to the simulation result without implementing any suggested control measures, implementing protection measures and smoking only addiction and dual addictions improvement (rehabilitation) control measures simultaneously (i.e., implementing Strategy G) has a significant role to reduce the dual addiction dissemination in the community. Similarly, the simulation curve shown in Fig 8(H) shows the potential impacts (effects) of concurrently implementing alcoholism only and dual addictions improvement (rehabilitation) control measures (i.e., implementing Strategy H) on the problem of dual addiction dissemination in the community. The simulation result presented in Fig 8(H) demonstrates that putting protective measures in place concurrently with alcoholism-only and multiple addictions improvement (rehabilitation) management measures (i.e., applying Strategy H) has a significant influence in reducing the spread of dual addictions. Lastly, let's compare the numerical simulation results shown in Fig 8F–8H without doing the cost-effectiveness analysis. We find that Strategy H is the most successful tactic used by the stakeholders to lessen the problem of alcoholism and smoking dual addiction spreading throughout the community.

**5.3.4 Effect of all control measures on the dual addiction.** Here, the numerical simulation result described by Fig 9 reveals the effects (impacts) of implementing all the smoking addiction, alcoholism addiction, and smoking and alcoholism dual addiction protection and improvement (rehabilitation) control measures ($w_1 \neq 0$, $w_2 \neq 0$, $w_3 \neq 0$, $w_4 \neq 0$, $w_5 \neq 0$, $w_6 \neq 0$) simultaneously (i.e., implementing the Strategy I) on the dual addiction dissemination problem in the community. From the result of the simulation illustrated by Fig 9 one can observe that implementing Strategy I has a fundamental impact to reduce the dual addiction dissemination problem in the community as compared with all the simulation cures from Figs 4–8

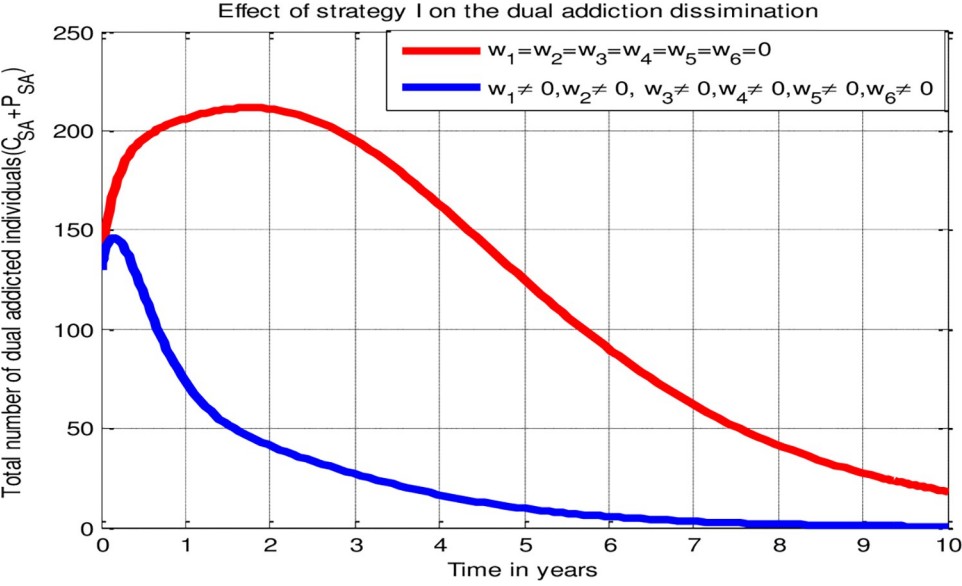

**Fig 9. Impact of Strategy I on the dual addiction dissemination.**

discussed previously and the simulation results obtained without implementing any proposed control measures. Therefore, without considering the cost-effectiveness analysis of the nine proposed control measures discussed above, Strategy I, is the most influential strategy to reduce the dual addiction dissemination problem in the community. Thus, the public stake holders shall put their effort to implement this strategy to reduce the dual addiction dissemination in the community.

## 6. Analyses of cost-effectiveness

To minimize, bring under control, or possibly eliminate the smoking and alcoholism dual addiction dissemination problem in the community, a significant amount of money, time, or both could be taken. This makes such a cost-effectiveness analysis very essential. Using comparable criteria to those found in [27], we conducted a cost-effectiveness analysis in this section to look at the expenses associated with rehabilitation control measures or protection control measures. In order to assess the cost-effectiveness of the control techniques we suggested, we also took into account the incremental cost-effectiveness ratio (ICER) approach. Comparing the costs and advantages of obtaining from the two distinct control measures is the aim of the ICER methodology. It is the ratio of the total number of instances prevented to the difference in expenditures incurred by two different techniques. By definition, the ICER is:

$$\text{ICER} = \frac{\text{Change in total costs in strategies A and B}}{\text{Change in control benefits in strategies A and B}},$$

where the differences in addiction avoided costs, expenses of protected cases, and rehabilitation costs are included in the ICER numerator, among other things. While the overall number of addictions avoided or the entire number of susceptibility cases safeguarded is taken into consideration by the ICER denominator when calculating variations in health outcomes. The cost-efficiency of various addiction control approaches is assessed using a set of criteria that include ranking the control strategies according to increasing efficacy in terms of the number of addicts prevented, with the strategy with the highest ICER value being considered the most effective. As a result, we conducted the cost-effectiveness analysis in this area of the research using the numerical simulations of the optimum control issue (15) that are completed in the sub-section 5.3.

### 6.1 Analysis of cost-effectiveness for case 1

where

$$\text{ICER (Strategy B)} = \frac{2.1716 \times 10^5 - 0}{4.3655 \times 10^6 - 0} = 4.97 \times 10^{-1},$$

$$\text{ICER (Strategy A)} = \frac{2.6261 \times 10^5 - 2.1716 \times 10^5}{4.6790 \times 10^6 - 4.3655 \times 10^6} = 4.638 \times 10^{-1},$$

From Table 5, we observed that ICER of strategy B has a higher value than Strategy A, it means that Strategy B is significantly more impracticable and prohibitively more expensive

**Table 5. Strategies, in ascending order of the total number of addictions that they avoided for case 1 (implementation of three strategies simultaneously).**

| Strategy | Total Addictions Averted | Total Cost Incurred ($) | ICER |
| --- | --- | --- | --- |
| B: $w_4, w_5, w_6 \neq 0$, $w_1 = w_2 = w_3 = 0$ | $4.3655 \times 10^6$ | $2.1716 \times 10^5$ | $4.97 \times 10^{-1}$ |
| A: $w_1, w_2, w_3 \neq 0$, $w_4 = w_5 = w_6 = 0$ | $4.6790 \times 10^6$ | $2.6261 \times 10^5$ | $4.638 \times 10^{-1}$ |

**Table 6. Strategies, in ascending order of the total number of addictions that they avoided for case 1 (implementation of four strategies simultaneously).**

| Strategy | Total Addictions Averted | Total Cost Incurred ($) | ICER |
|---|---|---|---|
| E: $w_1, w_2, w_3, w_6 \neq 0, w_4 = w_5 = 0$ | $1.675 \times 10^8$ | $5.262 \times 10^7$ | $3.142 \times 10^{-1}$ |
| C: $w_1, w_2, w_3, w_4 \neq 0, w_5 = w_6 = 0$ | $2.618 \times 10^8$ | $5.559 \times 10^7$ | $3.140 \times 10^{-2}$ |
| D: $w_1, w_2, w_3, w_5 \neq 0, w_4 = w_6 = 0$ | $4.897 \times 10^8$ | $3.882 \times 10^7$ | $-7.358 \times 10^{-2}$ |

than Strategy A. The results of the cost-effectiveness analysis illustrated in Table 5 reveal the ICER values for the two different strategies described in case 1(implementing the three control measures ($w_1, w_2, w_3 \neq 0, w_4 = w_5 = w_6 = 0$) simultaneously and from the result one can conclude that Strategy A is the most cost-effective strategy of minimizing the dissemination of smoking and alcoholism dual addiction in the community. Therefore, we recommend to the stakeholders that implementing all protection control measures simultaneously is the most cost-effective control strategy in case of implementing any three control measures simultaneously.

## 6.2 Analysis of cost-effectiveness for case 2

Incremental Cost-Effectiveness Ratio (ICER: Initially, let us compare the strategies E, C and D using their ICER ratio values that are computed below.

Here, using the ICER approach we computed the values illustrated in Table 6 and from the result we observed that the ICER value of Strategy E is dominant in terms of cost as compared with both Strategy C and D. Therefore, we have to removed Strategy E and consider the other two strategies for the next computation.

ICER values stated in Table 7 are computed as:

ICER (Strategy C) = $\frac{5.559 \times 10^7 - 0}{2.618 \times 10^8 - 0} = 2.123 \times 10^{-1}$, and

ICER (Strategy D) = $\frac{3.882 \times 10^7 - 5.559 \times 10^7}{4.897 \times 10^8 - 2.618 \times 10^8} = -8.48 \times 10^{-2}$.

Now comparing the ICER values described in Table 7 above and we observed that Strategy C is dominant in terms of cost incurred means it is more expensive than the cost incurred for Strategy D and hence we should have to discarded Strategy C and implement Strategy D. Hence, from the cost-effectiveness analysis using ICER values for different proposed control strategies described in Case 2, we found Strategy D (implement four control measures i.e., $w_1, w_2, w_3, w_5 \neq 0, w_4 = w_6 = 0$ simultaneously) is the most cost-effective strategy used to reduce the smoking and alcoholism dual addiction dissemination in the community.

## 6.3 Analysis of cost-effectiveness for case 3

Incremental Cost-Effectiveness Ratio (ICER: Initially, let us compare the strategies H, G and F using their ICER ratio values that are computed below.

Here, using the ICER approach we computed the values illustrated in Table 8 and from the result we observed that the ICER value of Strategy G is dominant in terms of cost as compared with both Strategy H and F. Therefore, we have to removed Strategy G and consider the other two strategies for the next computation.

ICER values stated in Table 9 are computed as:

**Table 7. Strategies, in ascending order of the total number of addictions that they avoided for case 1 (implementation of four strategies simultaneously).**

| Strategy | Total Addictions Averted | Total Cost Incurred ($) | ICER |
|---|---|---|---|
| C: $w_1, w_2, w_3, w_4 \neq 0, w_5 = w_6 = 0$ | $2.618 \times 10^8$ | $5.559 \times 10^7$ | $2.123 \times 10^{-1}$ |
| D: $w_1, w_2, w_3, w_5 \neq 0, w_4 = w_6 = 0$ | $4.897 \times 10^8$ | $3.882 \times 10^7$ | $-8.48 \times 10^{-2}$ |

**Table 8. Strategies, in ascending order of the total number of addictions that they avoided for case 1 (implementation of five strategies simultaneously).**

| Strategy | Total Addictions Averted | Total Cost Incurred ($) | ICER |
|---|---|---|---|
| H: $w_1, w_2, w_3, w_5, w_6 \neq 0, w_4 = 0$ | $9.772 \times 10^6$ | $1.1559 \times 10^6$ | $1.1829 \times 10^{-1}$ |
| G: $w_1, w_2, w_3, w_4, w_6 \neq 0, w_5 = 0$ | $1.211 \times 10^7$ | $1.6415 \times 10^6$ | $6.5250 \times 10^{-1}$ |
| F: $w_1, w_2, w_3, w_4, w_5 \neq 0, w_6 = 0$ | $1.331 \times 10^7$ | $9.1930 \times 10^5$ | $-6.0183 \times 10^{-1}$ |

**Table 9. Strategies, in ascending order of the total number of addictions that they avoided for case 1 (implementation of five strategies simultaneously).**

| Strategy | Total Addictions Averted | Total Cost Incurred ($) | ICER |
|---|---|---|---|
| H: $w_1, w_2, w_3, w_5, w_6 \neq 0, w_4 = 0$ | $9.772 \times 10^6$ | $1.1559 \times 10^6$ | $1.1829 \times 10^{-1}$ |
| F: $w_1, w_2, w_3, w_4, w_5 \neq 0, w_6 = 0$ | $1.331 \times 10^7$ | $9.1930 \times 10^5$ | $-6.6874 \times 10^{-2}$ |

$$\text{ICER (Strategy H)} = \frac{1.1559 \times 10^6 - 0}{9.772 \times 10^6 - 0} = 1.1829 \times 10^{-1}, \text{ and}$$

$$\text{ICER (Strategy F)} = \frac{9.1930 \times 10^5 - 1.1559 \times 10^6}{1.331 \times 10^7 - 9.772 \times 10^6} = -6.6874 \times 10^{-2}.$$

Now comparing the ICER values described in Table 9 above and we observed that Strategy H is dominant in terms of cost incurred means it is more expensive than the cost incurred for Strategy F and hence we should have to discarded Strategy H and implement Strategy F. Hence, from the cost-effectiveness analysis using ICER values for different proposed control strategies described in Case 3, we found Strategy F (implementing five control measures i.e., $w_1, w_2, w_3, w_4, w_5 \neq 0, w_6 = 0$ simultaneously) is the most cost-effective strategy used to reduce the smoking and alcoholism dual addiction dissemination in the community.

### 6.4 Analysis of cost-effectiveness for case 4

Table 10 investigates the cost-effectiveness of implementing the Strategy I (the strategy described in Case 4) i.e., implemented all the proposed control measures, only one control strategy is viable. As a result, in Case 4, Strategy I is the most cost-effective method. Finally, we have collected the most economical strategy for each of the scenarios discussed in the above sub-sections that will effectively minimize the impact of dual addiction in the community. In order to do this, let us compare the most cost-effective strategy collected from each case.

### 6.5 The over all cost-effectiveness analysis

In this sub-section, we arranged the most cost-effective strategies that are implemented for each proposed cases illustrated above according to the ascending order of total avoided dual addiction cases which is described in Table 11 below.

From Table 11 above we observed that Strategy I has larger ICER value and hence it is the dominant strategy in terms of cost incurred. Thus, we removed Strategy I from Table 11 and compute the ICER values for the remaining three strategies as follows.

On Table 12 above we have computed the ICER values for Strategies A, F, and D and observed the result that Strategy I has larger ICER value and hence it is the dominant strategy in terms of cost incurred. Thus, we have to remove Strategy D from Table 12 and compute the ICER values for the remaining two strategies as follows.

**Table 10. Strategies, in ascending order of the total number of addictions that they avoided for Case 1 (implementation all proposed strategies simultaneously).**

| Strategy | Total Addictions Averted | Total Cost Incurred ($) | ICER |
|---|---|---|---|
| I: $w_1, w_2, w_3, w_4, w_5, w_6 = 0$ | $1.4354 \times 10^7$ | $1.9857 \times 10^7$ | $1.3834 \times 10^{-1}$ |

**Table 11. The collections of each of the most cost-effective strategy from each case with ascending order of overall avoided dual addictions cases.**

| Strategy | Total Addictions Averted | Total Cost Incurred ($) | ICER |
|---|---|---|---|
| A: $w_1, w_2, w_3 \neq 0, w_4 = w_5 = w_6 = 0$ | $4.6790 \times 10^6$ | $2.6261 \times 10^5$ | $5.6125 \times 10^{-2}$ |
| F: $w_1, w_2, w_3, w_4, w_5 \neq 0, w_6 = 0$ | $1.331 \times 10^7$ | $9.1930 \times 10^5$ | $7.6085 \times 10^{-2}$ |
| I: $w_1, w_2, w_3, w_4, w_5, w_6 \neq 0$ | $1.4354 \times 10^7$ | $1.9857 \times 10^6$ | $9.7899 \times 10^{-1}$ |
| D: $w_1, w_2, w_3, w_5 \neq 0, w_4 = w_6 = 0$ | $4.897 \times 10^8$ | $3.882 \times 10^7$ | $7.7489 \times 10^{-2}$ |

**Table 12. The collections of each of the most cost-effective strategy from Table 11 with ascending order of overall avoided dual addictions cases.**

| Strategy | Total Addictions Averted | Total Cost Incurred ($) | ICER |
|---|---|---|---|
| A: $w_1, w_2, w_3 \neq 0, w_4 = w_5 = w_6 = 0$ | $4.6790 \times 10^6$ | $2.6261 \times 10^5$ | $5.6125 \times 10^{-2}$ |
| F: $w_1, w_2, w_3, w_4, w_5 \neq 0, w_6 = 0$ | $1.331 \times 10^7$ | $9.1930 \times 10^5$ | $7.6085 \times 10^{-2}$ |
| D: $w_1, w_2, w_3, w_5 \neq 0, w_4 = w_6 = 0$ | $4.897 \times 10^8$ | $3.882 \times 10^7$ | $7.9558 \times 10^{-2}$ |

**Table 13. The collections of each of the most cost-effective strategy from Table 12 with ascending order of overall avoided dual addictions cases.**

| Strategy | Total Addictions Averted | Total Cost Incurred($) | ICER |
|---|---|---|---|
| A: $w_1, w_2, w_3 \neq 0, w_4 = w_5 = w_6 = 0$ | $4.6790 \times 10^6$ | $2.6261 \times 10^5$ | $5.6125 \times 10^{-2}$ |
| F: $w_1, w_2, w_3, w_4, w_5 \neq 0, w_6 = 0$ | $1.331 \times 10^7$ | $9.1930 \times 10^5$ | $7.6085 \times 10^{-2}$ |

On Table 13 above we have computed the ICER values for Strategies A and F and observed the result that Strategy F has larger ICER value and hence it is the dominant strategy in terms of cost incurred. According to the cost analyses results for the ICER values for various proposed strategies for all cases, we find that Strategy A is the most cost-effective control strategy that should be implemented to reduce the smoking and alcoholism dual addiction dissemination in the community. Therefore, we recommend for the public health stakeholders and policy makers to implement all the protection strategies ($w_1, w_2, w_3 \neq 0, w_4 = w_5 = w_6 = 0$) simultaneously, which is the most cost-effective control strategy.

## 7. Conclusions and future directions

In this study, we formulated the smoking and alcohol dual addiction dissemination model, calculated the sub-models and the dual addiction model effective reproduction numbers and the single addiction sub-models and dual addiction model free and dual addiction persistence equilibrium points. We proved the local stabilities of the addiction free equilibrium points and the phenomenon of backward bifurcation. The dual addiction model's optimality system, which establishes the conditions required enhancing smoking and alcohol dual addiction control, is constructed using Pontryagin's Maximum Principle. The control measures we considered include punish or/and educating individuals about smoking only addiction, alcohol only addiction and dual addiction dissemination problems ($w_1, w_2, w_3 \neq 0$) respectively, giving suitable treatments for smoking only addicted, alcohol only addicted and dual addicted individuals ($w_4, w_5, w_6 \neq 0$) respectively. To perform sensitivity analysis and simulate the optimal control problem we adopted parameter values from the available existing literatures, otherwise, we made suitable assumptions to carry out the simulation for illustration purposes. Moreover, we carried out the cost-effectiveness analysis of the proposed control strategies. The result of the cost-effectiveness analysis verifies that Strategy A is the most cost-effective strategy used to reduce the smoking and alcoholism dual addiction dissemination in the community. Thus, we recommend for the public health stakeholders and policy makers to give great

attention for the effort to implement Strategy A (implement all the protection control mechanisms) to reduce and eliminate the smoking and alcohol dual addiction dissemination dynamics in the community. In the future, potential researchers will make an effort to fit the model with the existing real data on the smoking and alcoholism dual addiction model, exploring the optimal control problem analysis of model by incorporating the age structure of individuals in the study, extending the model by including the stochastic approach or the fractional derivative approach or environmental and media factors.

## Author Contributions

**Conceptualization:** Shewafera Wondimagegnhu Teklu, Belela Samuel Kotola, Haileyesus Tessema Alemneh.

**Formal analysis:** Shewafera Wondimagegnhu Teklu, Belela Samuel Kotola, Haileyesus Tessema Alemneh.

**Investigation:** Shewafera Wondimagegnhu Teklu, Haileyesus Tessema Alemneh.

**Methodology:** Shewafera Wondimagegnhu Teklu, Belela Samuel Kotola, Haileyesus Tessema Alemneh.

**Resources:** Shewafera Wondimagegnhu Teklu.

**Validation:** Shewafera Wondimagegnhu Teklu.

**Visualization:** Shewafera Wondimagegnhu Teklu.

**Writing – original draft:** Shewafera Wondimagegnhu Teklu, Belela Samuel Kotola, Haileyesus Tessema Alemneh.

**Writing – review & editing:** Shewafera Wondimagegnhu Teklu, Belela Samuel Kotola, Haileyesus Tessema Alemneh.

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
