## [Decision Letter · Decision Letter 0]

4 Jul 2024

PONE-D-24-23154Modeling and Analysis of Smoking and Alcoholism Dual Addiction Dynamics with Optimal Control Theory and Cost-EffectivenessPLOS ONE

Dear Dr. Teklu,

Thank you for submitting your manuscript to PLOS ONE. After careful consideration, we feel that it has merit but does not fully meet PLOS ONE’s publication criteria as it currently stands. Therefore, we invite you to submit a revised version of the manuscript that addresses the points raised during the review process.

We look forward to receiving your revised manuscript.

Kind regards,

Joshua Kiddy K. Asamoah, PhD

Academic Editor

PLOS ONE

Journal Requirements:

Additional Editor Comments:

1. The equation after the statement "The sum of all the differential equations described in equation (4)" should be on a new line and numbered. 

2. Revise the abstract to contain the public health implications of the studies. 

3. Explain the terms in the basic reproduction number obtained, as done on page 6 of this paper https://doi.org/10.1016/j.heliyon.2023.e20531 after equation (5). Also, comment on the above paper since it is relevant to the current study. 

3. What informed the authors' choice of the control terms since the authors did not perform a sensitivity analysis of the parameters in the R0.

Reviewers' comments:

Reviewer's Responses to Questions

**Comments to the Author**

1. Is the manuscript technically sound, and do the data support the conclusions?

Reviewer #1: Yes

Reviewer #2: Yes

2. Has the statistical analysis been performed appropriately and rigorously? 

Reviewer #1: Yes

Reviewer #2: N/A

3. Have the authors made all data underlying the findings in their manuscript fully available?

Reviewer #1: Yes

Reviewer #2: Yes

4. Is the manuscript presented in an intelligible fashion and written in standard English?

Reviewer #1: No

Reviewer #2: Yes

5. Review Comments to the Author

Reviewer #1: The study seeks to formulate a mathematical model to help mitigate the societal addictions to both smoking and alcohol intake. I find the work interesting if improved. These are my comments to the authors.

1. The abstract of the study contains language construction errors. Also, the statement ". Finally, from the cost-effectiveness analyses we observed that

implementing strategy is the most cost-effective strategy." in the abstract is not clear.

2. Authors should desist from citations of the form [24-32] where many papers are cited without explicitly explaining each of them and their results. There are several of them, kindly cite relevant articles with their key results well explained.

3. There are several grammatical errors in the work. Kindly read through the entire paper and correct them.

4. Please reconsider this statement "In the best of our understanding organized from a thorough literature review, no one is studied

the smoking and alcoholism dual addiction dissemination in the community using mathematical

modeling approach. " From my little search i came across this paper "Bhunu, C. P., & Mushayabasa, S. (2012). A theoretical analysis of smoking and alcoholism. Journal of Mathematical Modelling and Algorithms, 11, 387-408." which has extensively studied alcohol and smoking dual usage.

5. Please change these phrases in the work "minimize and tackle".

6. Read the paper I have given out in point 4 and rewrite your contribution to literature. The novelty of the research is in doubt. Please clarify that. Why is the study not employing fractional calculus but integer order derivatives?

7. The section 2.1 to 2.3 headings are unnecessary. Let them all fall under section 2.

8. Under model formulation, the compartment named as protection definition is ambiguous. Kindly explain that well and further established that from literature.

9. Under model formulation, what do the authors want to communicate regarding the two compartments; addicted and permanently addicted? Please be clear with their definitions and support them with literature.

10. Kindly subject your assumptions to literature.

11. Under schematic diagram, why should a permanently addicted individual (to alcohol or smoking) become a dual addicted before becoming a permanently dual addicted to both?

12. the model failed to consider an alcohol related or smoking related or even alcohol-smoking related death rates. Why?

13. Authors should be explicit as in what it means that their model is positive and bounded.

14. Under the optimal control, authors should be specific with the efforts they are referring to. the controls look abstract in their current form.

15. Why is this "Characterization of Optimal Control" underlined? The paper has formatting problems. Kindly fix that.

16. A number of the model parameters are assumed. Will the model be able to stand the test of time?

17. The conclusion is too long. Paragraph one seems to be overelaborated. Other paragraphs should be simplified. Is there any literature support to your conclusions?

Reviewer #2: Review report on the manuscript “Modeling and Analysis of Smoking and Alcoholism Dual Addiction Dynamics with Optimal Control Theory and Cost-Effectiveness”

I have carefully read the paper, and it seems interesting. Therefore, the manuscript can be considered further for publication in PLOS ONE if the authors are willing to incorporate a MAJOR Revision. Here under are my specific comments on the manuscript:

1. It is strongly advised and it is the responsibility of the authors to check the whole manuscript and correct all the grammatical errors and typos. The manuscript is full of grammatical errors and typos. It is enough to recommend rejection if the authors fail to thoroughly proofread their revised manuscript.

2. In the abstract, it is not clear on what control interventions were the optimal strategies and cost-effectiveness analysis carried on.

3. The last sentence in the abstract should be split into simple sentences with clear information.

4. There are a number of compound sentences with no clear meanings throughout the manuscript. For instance, see the first sentence under the introduction section. I suggest that the authors split this kind of sentences into simple ones with clear meanings.

5. There are a number of long sentences in the paper. For instance, the first sentence in the first paragraph on the second page is too long. The sentence should be split into simple and meaningful sentences.

6. The literature review should be updated with the following related works:

a) https://doi.org/10.1007/s40314-022-01760-2

b) https://www.lhscientificpublishing.com/Journals/articles/DOI-10.5890-JAND.2023.03.004.aspx

7. Description of the parameters delta_1, delta_2, delta_3, rho_1, rho_2 and rho_3 being "modification parameters" as mentioned by the authors in Table 1 is not clear. The physical meanings of the parameters should be given.

8. theta_1 and theta_2 should be described better than just being modification parameters. Please, see the recommended articles in (6) for similar parameter descriptions.

9. In the flowchart in Figure 1, the rate of progression from P_S to C_SA is missing. The authors should fix this.

10. The adjoint variables f_i,i=1,2,...,11 in Equation (21) and thereafter are different from their original definitions (with boldface). The authors should make corrections accordingly.

11. What is the meaning of (.) as appeared in the optimal state variables S, P and others in Theorem 9?

12. Remove the full stop (.) from the caption of all the figures.

13. Revise the captions of Figures 5 to 8 to reflect the specific strategy in terms of Strategy A to Strategy I.

6. PLOS authors have the option to publish the peer review history of their article (what does this mean?). If published, this will include your full peer review and any attached files.

Reviewer #1: No

Reviewer #2: No

---

## [Author Response · Author response to Decision Letter 0]

16 Jul 2024

Dear Editor, we have revised our manuscript “Smoking and Alcoholism Dual Addiction Dissemination Model Analysis with Optimal Control Theory and Cost-Effectiveness” accordingly.

Editor Comments and Authors Responses

https://journals.plos.org/plosone/s/file?id=wjVg/PLOSOne-formatting sample main body.pdf and https://journals.plos.org/plosone/s/file?id=ba62/PLOSOne-formatting sample title authors affiliations.pdf

Response: We have revised according to our previous experience in PLOS ONE Journal.

 Please note that PLOS ONE has specific guidelines on code sharing for submissions in which author-generated code underpins the findings in the manuscript. In these cases, we expect all author-generated code to be made available without restrictions upon publication of the work. Please review our guidelines at https://journals.plos.org/plosone/s/materials-and-software-sharing#loc-sharing-code and ensure that your code is shared in a way that follows best practice and facilitates reproducibility and reuse.

Response: We agree with this comment 

 We note that your Data Availability Statement is currently as follows: All relevant data are within the manuscript and its Supporting Information files.

Response: Since we collected parameter values from published literatures and assumed values for illustration purposes, we mentioned “All relevant data are within the manuscript and its Supporting Information files” in the revised manuscript.

Authors do not need to submit their entire data set if only a portion of the data was used in the reported study. If your submission does not contain these data, please either upload them as Supporting Information files or deposit them to a stable, public repository and provide us with the relevant URLs, DOIs, or accession numbers. For a list of recommended repositories, please see https://journals.plos.org/plosone/s/recommended-repositories.

Response: All relevant data are within the manuscript and its Supporting Information files. 

Additional Editor Comments:

1. The equation after the statement "The sum of all the differential equations described in equation (4)" should be on a new line and numbered. 

Response: We have corrected accordingly. 

2. Revise the abstract to contain the public health implications of the studies. 

Response: Since both smoking and alcohol addictions are related to health issues implementing the Strategy A has a great health implication regarding to reduce the dual addiction dissemination problem in the community. Accordingly we have revised the abstract.

3. Explain the terms in the basic reproduction number obtained, as done on page 6 of this paper https://doi.org/10.1016/j.heliyon.2023.e20531 after equation (5). Also, comment on the above paper since it is relevant to the current study. 

Response: We strongly agree with this strong comment and revised our manuscript accordingly by cited this up-to-date and related paper. See the last part of the sub-section 3.3.2.

4. What informed the authors' choice of the control terms since the authors did not perform a sensitivity analysis of the parameters in the R0.

Response: In analysis of our initial manuscript we performed sensitivity analysis but the total number of pages for the manuscript was more than 46 pages due to that we did not incorporated sensitivity analysis. Now since the editor comment 4 is very crucial in the revised manuscript based on the model parameters we have performed the sensitivity analysis together with its associated diagram and justified the corresponding optimal control problem.

Reviewer #1: The study seeks to formulate a mathematical model to help mitigate the societal addictions to both smoking and alcohol intake. I find the work interesting if improved. These are my comments to the authors.

1. The abstract of the study contains language construction errors. Also, the statement ". Finally, from the cost-effectiveness analyses we observed that implementing strategy is the most cost-effective strategy." in the abstract is not clear.

Response: We revised the abstract accordingly.

2. Authors should desist from citations of the form [24-32] where many papers are cited without explicitly explaining each of them and their results. There are several of them, kindly cite relevant articles with their key results well explained.

Response: We have restructured the literatures accordingly.

3. There are several grammatical errors in the work. Kindly read through the entire paper and correct them.

Response: We agree with this basic comment and we revised the full manuscript accordingly.

4. Please reconsider this statement "In the best of our understanding organized from a thorough literature review, no one is studied the smoking and alcoholism dual addiction dissemination in the community using mathematical modeling approach.” From my little search i came across this paper "Bhunu, C. P., & Mushayabasa, S. (2012). A theoretical analysis of smoking and alcoholism. Journal of Mathematical Modelling and Algorithms, 11, 387-408, which has extensively studied alcohol and smoking dual usage.

Response: We appreciate the reviewer search for similar studies what we have not addressed initially, however, after considering the reviewer comment we totally agree with the comment and searched to get similar studies in the same area thorough literature review and we could not find a study that have been constructed and analyzed the smoking and alcoholism dual addiction dynamics in the community with optimal control strategies and cost-effectiveness. Therefore, our study is different from the study suggested by the reviewer in different aspects.

5. Please change these phrases in the work "minimize and tackle".

Response: We have corrected accordingly.

6. Read the paper I have given out in point 4 and rewrite your contribution to literature. The novelty of the research is in doubt. Please clarify that. Why is the study not employing fractional calculus but integer order derivatives?

Response: We strongly agree and we have re-written the contribution and novelty of our study by comparing it with other similar studies like mentioned in comment 4. In our study we constructed the integer order model and in the near future any researcher can modify our study by considering fractional order approach to verify the effect of memory.

7. The section 2.1 to 2.3 headings are unnecessary. Let them all fall under section 2.

Response: We have ignore all the mentioned section headings accordingly.

8. Under model formulation, the compartment named as protection definition is ambiguous. Kindly explain that well and further established that from literature.

Response: The compartment P(t) contains individuals who are protected against either smoking addiction (by education) or alcohol addiction (by education) or dual addiction denoted by P(t). In similar approaches used in the studies of real world situations stated in [30, 31, 32] we considered education as the prevention mechanism.

9. Under model formulation, what do the authors want to communicate regarding the two compartments; addicted and permanently addicted? Please be clear with their definitions and support them with literature.

Response: We considered individuals who are addicted with smoking, but, can be improved (rehabilitated by taking treatment) denoted by〖 I〗_S (t), individuals who are permanently addicted with smoking throughout their life (i.e., could not be rehabilitated by taking any treatment measures) denoted by 〖 P〗_S (t), individuals who are addicted with alcohol, but, can be improved (rehabilitated by taking treatment) denoted by〖 I〗_A (t), individuals who are permanently addicted with alcohol throughout their life (i.e., could not be rehabilitated by taking any treatment measures) denoted by 〖 P〗_A (t). Since anxiety against mathematics is a serious social problem we refer for the case of permanent addiction from permanent anxiety from literature [29].

10. Kindly subject your assumptions to literature.

Response: Based on real world problem studies the studies stated in literatures [30, 31, 32] we have revised our model descriptions and assumptions accordingly.

11. Under schematic diagram, why should a permanently addicted individual (to alcohol or smoking) become a dual addicted before becoming a permanently dual addicted to both?

Response: We agree with this comment since both options are possible. In our model descriptions and assumptions we considered that individuals in the compartment 〖 C〗_SA are dually addicted but they may be either smoking or alcohol permanent addicted. But, individuals in the compartment 〖 P〗_SA are all permanently dual addicted with both smoking and alcohol. Thus, even though, there is a possibility where a permanently addicted individual (to alcohol or smoking) become a permanently dual addicted to both for the sake of simplicity we assumed that individuals who are either smoking or alcohol permanently addicted will be dually addicted before permanently dually addicted. 

12. the model failed to consider an alcohol related or smoking related or even alcohol-smoking related death rates. Why?

Response: Initially we ignore death rate with addictions, however, after considering the reviewer comments we incorporated death rates due to illness related with addictions. Therefore, the revised manuscript considers different death rates due to illnesses related with addictions.

13. Authors should be explicit as in what it means that their model is positive and bounded.

Response: The model (4) solutions are both positive and bounded in the region Ω={█((S,P,〖 E〗_S,〖 I〗_S,〖 P〗_S,〖 E〗_A,〖 I〗_A,P_A,C_SA,P_SA,T)∈R_+^11,N(t)≤Κ/μ)} means it is invariant and attracting for dual addiction dynamical system (4). Thus, the dual addiction model is mathematically and epidemiologically well-posed and it is sufficient to consider the dynamics of the flow generated by the system (4) in Ω [28, 43].

14. Under the optimal control, authors should be specific with the efforts they are referring to. the controls look abstract in their current form.

Response: We have considered specific control measures such as education, punishment and psychological treatments.

15. Why is this "Characterization of Optimal Control" underlined? The paper has formatting problems. Kindly fix that.

Response: We have addressed this comment accordingly.

16. A number of the model parameters are assumed. Will the model be able to stand the test of time?

Response: In this study, to perform sensitivity analysis and to simulate the optimal control problem we adopted most of the parameter values from the available existing literatures, otherwise, we made suitable assumptions to carry out the simulations for illustration purposes.

17. The conclusion is too long. Paragraph one seems to be overelaborated. Other paragraphs should be simplified. Is there any literature support to your conclusions?

Response: We have revised the whole conclusion part very carefully.

Reviewer #2: Review report on the manuscript “Modeling and Analysis of Smoking and Alcoholism Dual Addiction Dynamics with Optimal Control Theory and Cost-Effectiveness”

I have carefully read the paper, and it seems interesting. Therefore, the manuscript can be considered further for publication in PLOS ONE if the authors are willing to incorporate a MAJOR Revision. Here under are my specific comments on the manuscript:

1. It is strongly advised and it is the responsibility of the authors to check the whole manuscript and correct all the grammatical errors and typos. The manuscript is full of grammatical errors and typos. It is enough to recommend rejection if the authors fail to thoroughly proofread their revised manuscript.

Response: We agree with this basic comment and we revised the full manuscript accordingly.

2. In the abstract, it is not clear on what control interventions were the optimal strategies and cost-effectiveness analysis carried on.

Response: The study carried out cost-effectiveness analysis to compare the cost incurred when we applied each of the proposed strategies A to I and in the revised manuscript abstract section we explained as “The cost-effectiveness analysis shows that implementing all the protection (education) control measures simultaneously (i.e. implementing Strategy A) is the most cost-effective strategy”

3. The last sentence in the abstract should be split into simple sentences with clear information.

Response: We re-wrote it again accordingly.

4. There are a number of compound sentences with no clear meanings throughout the manuscript. For instance, see the first sentence under the introduction section. I suggest that the authors split this kind of sentences into simple ones with clear meanings.

Response: We revised accordingly.

5. There are a number of long sentences in the paper. For instance, the first sentence in the first paragraph on the second page is too long. The sentence should be split into simple and meaningful sentences.

Response: We revised accordingly.

6. The literature review should be updated with the following related works:

a) https://doi.org/10.1007/s40314-022-01760-2

b) https://www.lhscientificpublishing.com/Journals/articles/DOI-10.5890-JAND.2023.03.004.aspx

Response: We have incorporated these up-to-date and related peer-reviewed published literatures to improve our revised manuscript.

7. Description of the parameters delta_1, delta_2, delta_3, rho_1, rho_2 and rho_3 being "modification parameters" as mentioned by the authors in Table 1 is not clear. The physical meanings of the parameters should be given.

Response: We have given the right physical meanings of all the mentioned parameters in the revised manuscript accordingly.

8. theta_1 and theta_2 should be described better than just being modification parameters. Please, see the recommended articles in (6) for similar parameter descriptions.

Response: We mean that θ_1 is the modification parameter that verify smoking and alcohol dual addicted individuals are more involved in the alcohol addiction dissemination process than the alcohol only addicted individuals. Similarly, we mean that θ_2 is the modification parameter that verify smoking and alcohol dual addicted individuals are more involved in the smoking addiction dissemination process than the smoking only addicted individuals.

9. In the flowchart in Figure 1, the rate of progression from P_S to C_SA is missing. The authors should fix this.

Response: Sorry for our mistake not to incorporate the term ρ_2 λ_A in the model flow chart what we have considered in the derivative of the state variables P_S and C_(SA ) for the dynamical system (4). In the revised manuscript we have corrected this mistake.

10. The adjoint variables f_i,i=1,2,...,11 in Equation (21) and thereafter are different from their original definitions (with boldface). The authors should make corrections accordingly.

Response: We have corrected accordingly.

11. What is th

---

## [Decision Letter · Decision Letter 1]

26 Jul 2024

PONE-D-24-23154R1Smoking and Alcoholism Dual Addiction Dissemination Model Analysis with Optimal Control Theory and Cost-EffectivenessPLOS ONE

Dear Dr. Teklu,

Thank you for submitting your manuscript to PLOS ONE. After careful consideration, we feel that it has merit but does not fully meet PLOS ONE’s publication criteria as it currently stands. Therefore, we invite you to submit a revised version of the manuscript that addresses the points raised during the review process.

**ACADEMIC EDITOR:**

General comment

The abstract and conclusion are well-written. However, the keywords should not contain words already in the title.

Other comments.

(1) The manuscript has a few grammatical errors and typos. The authors need to rectify these.

(2) Please clearly state the novelty of the work in connection to the mathematical modelling of the disease. What new compartment(s) have the authors added, and why do they need to add those compartment(s)? 

(3) The authors should indicate in their motivation the new mathematical analysis used in studying this dynamics, which other researchers haven’t considered.

(4) In the numerical section, the authors should improve on discussing the various Figures obtained and their impact on smoking and alcoholism control.

**I kindly request you to provide a thoroughly revised manuscript along with a point-by-point response delineating how you addressed my general and specific comments; I meant exactly that. As to the point-by-point response, revised text in different colours is necessary, but that is not sufficient to let me know that you modified any given issue; it is necessary to answer specifically HOW you addressed/changed each issue in the response letter. Please understand that I must request another revision/rejection of your manuscript if the revisions are inadequate.

We look forward to receiving your revised manuscript.

Kind regards,

Joshua Kiddy K. Asamoah, PhD

Academic Editor

PLOS ONE

Journal Requirements:

Reviewers' comments:

Reviewer's Responses to Questions

**Comments to the Author**

1. If the authors have adequately addressed your comments raised in a previous round of review and you feel that this manuscript is now acceptable for publication, you may indicate that here to bypass the “Comments to the Author” section, enter your conflict of interest statement in the “Confidential to Editor” section, and submit your "Accept" recommendation.

Reviewer #1: All comments have been addressed

Reviewer #2: All comments have been addressed

2. Is the manuscript technically sound, and do the data support the conclusions?

Reviewer #1: Yes

Reviewer #2: Yes

3. Has the statistical analysis been performed appropriately and rigorously? 

Reviewer #1: N/A

Reviewer #2: N/A

4. Have the authors made all data underlying the findings in their manuscript fully available?

Reviewer #1: Yes

Reviewer #2: Yes

5. Is the manuscript presented in an intelligible fashion and written in standard English?

Reviewer #1: Yes

Reviewer #2: Yes

6. Review Comments to the Author

Reviewer #1: The authors have responded to many of my comments accurately. Notwithstanding this, kindly check these comments to improve the paper.

1. The paper still has some grammatical errors. Check the last statement in the abstract and also the entire paper.

2. The literature support for the assumptions should be cited before stating the assumptions.

3. Regarding my comment 8, your response indicated that similar works are done in [30-32], but you cited [38-40] instead in the article. Which one is the proper citation?

4. The manuscript still has a lot of formatting inaccuracies. Please work on them.

Reviewer #2: The authors have carefully addressed all my comments to satisfaction. Thus, the manuscript can be considered further for publication by Plos One.

7. PLOS authors have the option to publish the peer review history of their article (what does this mean?). If published, this will include your full peer review and any attached files.

Reviewer #1: No

Reviewer #2: **Yes: **Afeez Abidemi

---

## [Author Response · Author response to Decision Letter 1]

2 Aug 2024

Dear Editor, we the authors of the manuscript entitled “Smoking and Alcoholism Dual Addiction Dissemination Model Analysis with Optimal Control Theory and Cost-Effectiveness” thanks all the editors and reviewers for their great contribution in the review process and sharing knowledge of real world problem dynamics. Dear, Sir we have the following point by point responses.

Editor and Reviewers Comments and Authors Responses

Editor Comments and Authors Responses

The abstract and conclusion are well-written. However, the keywords should not contain words already in the title.

Response: According to the comment given above, we removed some word from the previous manuscript and the keywords in the revised manuscript include: Smoking; Alcoholism; Smoking and Alcoholism Dual Addiction; Optimal Control Theory; Cost-Effectiveness, all are already in the title of the revised manuscript.

Other comments:

 The manuscript has a few grammatical errors and typos. The authors need to rectify these.

Response: According to the comment given to us we have read the whole manuscript again and again and corrected some grammatical errors we addressed in each page of the previous manuscript.

 Please clearly state the novelty of the work in connection to the mathematical modelling of the disease. What new compartment(s) have the authors added, and why do they need to add those compartment(s)? 

Response: Dear editor, whenever we the authors studied different co-infection models on infectious diseases and we reviewed various published sources carried out by different scholars on the transmission dynamics of co-infections of infectious diseases. Also, in the formulation of this proposed dual addiction model we reviewed different co-existence models, however, to the best of the our understanding no one is formulated and analyzed eleven compartmental co-infection model by considering a protection compartment against both infections with optimal control theory and cost-effectiveness analysis. In comparisons with other scholars’ co-existence (dual existence) mathematical modeling studies on infectious diseases or social behaviors the new compartments that incorporated in our formulated model are the protection compartment denoted by P and the compartment that contains individuals who are permanently dual addicted with both smoking and alcohol denoted by P_SA. We add these two compartments in our formulated model: 

 To investigate the impacts of pre-exposure mechanism such as the impact of education protection mechanism on the dual addiction dissemination dynamics in the community.

 To understand the smoking and alcoholism dual addicted individuals behavior, the dual addiction dissemination health impact, economic impact in the community and to verify how these addictions have long run impact on individuals who are permanently dual addicted with both smoking and alcohol ( since they cannot rehabilitated from their dual addiction). 

 The authors should indicate in their motivation the new mathematical analysis used in studying this dynamics, which other researchers haven’t considered.

Response: From response of the comment 3 above we have explained that no alcohol addiction or smoking addiction or smoking and alcohol dual addiction mathematical model studies incorporated the protected group (protected compartment) against both addictions (by education), the permanently dual addicted group, optimal control theory and cost-effectiveness analysis in the dissemination dynamics of these addictions. Similarly, in case of mathematical analysis no smoking and alcoholism dual addiction mathematical modeling research studies analyzed the phenomenon of backward bifurcation. 

 In the numerical section, the authors should improve on discussing the various Figures obtained and their impact on smoking and alcoholism control.

Response: We have improved the discussions by showing the impacts of the proposed control measures by comparing with other strategies in each case and the result determined without applying any control strategy.

**I kindly request you to provide a thoroughly revised manuscript along with a point-by-point response delineating how you addressed my general and specific comments; I meant exactly that. As to the point-by-point response, revised text in different colors is necessary, but that is not sufficient to let me know that you modified any given issue; it is necessary to answer specifically HOW you addressed/changed each issue in the response letter. 

Response: We revised according to the editor suggestions. 

Reviewer 1 Comments and Authors Responses

 The paper still has some grammatical errors. Check the last statement in the abstract and also the entire paper.

Response: We have corrected some grammatical errors we addressed in each page of the previous manuscript.

 The literature support for the assumptions should be cited before stating the assumptions.

Response: In page 5 of this revised manuscript, we made corrections and cited relevant studies for model assumptions before we started the assumptions. 

 Regarding my comment 8, your response indicated that similar works are done in [30-32], but you cited [38-40] instead in the article. Which one is the proper citation?

Response: Sorry for our mistake what we have done in the first revision process the according to the new reference arrangement in the first revision process the corrected citation should be [38-40]. Dear Sir, please consider the proper citation [38-41] for this revised manuscript. 

 The manuscript still has a lot of formatting inaccuracies. Please work on them.

Response: In the revised manuscript we have corrected various formatting inaccuracies

---

## [Decision Letter · Decision Letter 2]

12 Aug 2024

Smoking and Alcoholism Dual Addiction Dissemination Model Analysis with Optimal Control Theory and Cost-Effectiveness

PONE-D-24-23154R2

Dear Dr. Teklu,

We’re pleased to inform you that your manuscript has been judged scientifically suitable for publication and will be formally accepted for publication once it meets all outstanding technical requirements.

Kind regards,

Joshua Kiddy K. Asamoah, PhD

Academic Editor

PLOS ONE

Additional Editor Comments (optional):

Reviewers' comments:

Reviewer's Responses to Questions

**Comments to the Author**

1. If the authors have adequately addressed your comments raised in a previous round of review and you feel that this manuscript is now acceptable for publication, you may indicate that here to bypass the “Comments to the Author” section, enter your conflict of interest statement in the “Confidential to Editor” section, and submit your "Accept" recommendation.

Reviewer #1: (No Response)

2. Is the manuscript technically sound, and do the data support the conclusions?

Reviewer #1: Yes

3. Has the statistical analysis been performed appropriately and rigorously? 

Reviewer #1: N/A

4. Have the authors made all data underlying the findings in their manuscript fully available?

Reviewer #1: Yes

5. Is the manuscript presented in an intelligible fashion and written in standard English?

Reviewer #1: Yes

6. Review Comments to the Author

Reviewer #1: The paper is currently in a good state to be accepted for publication subject to these considerations.

1. In the abstract, authors used the word "created". I suggest that word be changed since all mathematical models are modifications or extensions based on the existing SIR model propounded by Kermack and McKendrick in 1927.

2. The references are not meeting the standards of Plos One journal. This should be done.

7. PLOS authors have the option to publish the peer review history of their article (what does this mean?). If published, this will include your full peer review and any attached files.

Reviewer #1: No

---

## [Editor Report · Acceptance letter]

19 Aug 2024

PONE-D-24-23154R2 

PLOS ONE

Dear Dr. Teklu, 

I'm pleased to inform you that your manuscript has been deemed suitable for publication in PLOS ONE. Congratulations! Your manuscript is now being handed over to our production team.

Kind regards, 

on behalf of

Dr. Joshua Kiddy K. Asamoah 

Academic Editor

PLOS ONE